# Hippocampal and cortical mechanisms at retrieval explain variability in episodic remembering in older adults

Alexandra N Trelle[1]*, Valerie A Carr[1†], Scott A Guerin[1], Monica K Thieu[1‡], Manasi Jayakumar[1‡], Wanjia Guo[1§], Ayesha Nadiadwala[1#], Nicole K Corso[1], Madison P Hunt[1], Celia P Litovsky[1¶], Natalie J Tanner[1], Gayle K Deutsch[2], Jeffrey D Bernstein[1**], Marc B Harrison[1], Anna M Khazenzon[1], Jiefeng Jiang[1††], Sharon J Sha[2], Carolyn A Fredericks[2‡‡], Brian K Rutt[3], Elizabeth C Mormino[2], Geoffrey A Kerchner[2], Anthony D Wagner[1]*

[1]Department of Psychology, Stanford University, Stanford, United States; [2]Department of Neurology & Neurological Sciences, Stanford University, Stanford, United States; [3]Department of Radiology & Radiological Sciences, Stanford University, Stanford, United States

*For correspondence:
atrelle@stanford.edu (ANT);
awagner@stanford.edu (ADW)

Present address: [†]San Jose State University, San Jose, United States; [‡]Columbia University, New York, United States; [§]University of Oregon, Eugene, United States; [#]University of Texas at Austin, Austin, United States; [¶]Johns Hopkins University, Baltimore, United States; [**]UC San Diego Medical School, San Diego, United States; [††]University of Iowa, Iowa City, United States; [‡‡]Yale University, New Haven, United States

Competing interests: The authors declare that no competing interests exist.

**Abstract** Age-related episodic memory decline is characterized by striking heterogeneity across individuals. Hippocampal pattern completion is a fundamental process supporting episodic memory. Yet, the degree to which this mechanism is impaired with age, and contributes to variability in episodic memory, remains unclear. We combine univariate and multivariate analyses of fMRI data from a large cohort of cognitively normal older adults (N=100) to measure hippocampal activity and cortical reinstatement during retrieval of trial-unique associations. Trial-wise analyses revealed that (a) hippocampal activity scaled with reinstatement strength, (b) cortical reinstatement partially mediated the relationship between hippocampal activity and associative retrieval, (c) older age weakened cortical reinstatement and its relationship to memory behaviour. Moreover, individual differences in the strength of hippocampal activity and cortical reinstatement explained unique variance in performance across multiple assays of episodic memory. These results indicate that fMRI indices of hippocampal pattern completion explain within- and across-individual memory variability in older adults.

## Introduction

Episodic memory – in particular the ability to form and retrieve associations between multiple event elements that comprise past experiences – declines with age (*Spencer and Raz, 1995*; *Rönnlund et al., 2005*; *Old and Naveh-Benjamin, 2008*). Retrieval of an episodic memory relies critically on hippocampal-dependent pattern completion, which entails reactivation of a stored memory trace by the hippocampus in response to a partial cue, leading to replay of cortical activity patterns that were present at the time of memory encoding (*Marr, 1971*; *McClelland et al., 1995*; *Tanaka et al., 2014*; *Staresina et al., 2019*). Given observed links between in vivo measures of pattern completion and episodic remembering (*Nakazawa et al., 2002*; *Gelbard-Sagiv et al., 2008*; *Gordon et al., 2014*), and evidence of altered hippocampal function with age (*Lister and Barnes, 2009*; *Leal and Yassa, 2013*), changes in hippocampal pattern completion may play an important role in explaining age-related impairments in episodic memory. While a leading hypothesis, the degree to which the integrity of pattern completion can explain (a) trial-to-trial differences in episodic remembering within older adults and (b) differences in memory performance between older individuals remain underspecified.

Functional MRI (fMRI) studies in younger adults suggest that hippocampal pattern completion is associated with at least two key neural markers: (a) an increase in hippocampal univariate activity (*Eldridge et al., 2000*; *Dobbins et al., 2003*; *Yonelinas et al., 2005*) and (b) cortical reinstatement of content-specific activity patterns present during encoding (*Nyberg et al., 2000*; *Wheeler et al., 2000*; *Kahn et al., 2004*). Multivariate pattern analyses — machine learning classification (*Norman et al., 2006*) and pattern similarity (*Kriegeskorte et al., 2008*) — reveal evidence for cortical reinstatement of categorical event features (*Polyn et al., 2005*; *Johnson and Rugg, 2007*; *Gordon et al., 2014*) and event-specific details (*Staresina et al., 2012*; *Ritchey et al., 2013*; *Kuhl and Chun, 2014*) during successful recollection. Moreover, hippocampal and cortical metrics of pattern completion covary, such that trial-wise fluctuations in hippocampal univariate retrieval activity are related to the strength of cortical reinstatement (*Staresina et al., 2012*; *Ritchey et al., 2013*; *Gordon et al., 2014*), and both hippocampal activity and reinstatement strength are related to associative retrieval performance (*Gordon et al., 2014*; *Gagnon et al., 2019*). These findings support models (*Marr, 1971*; *McClelland et al., 1995*; *Tanaka et al., 2014*) positing that cortical reinstatement depends, in part, on hippocampal processes, and contributes to remembering.

Initial data bearing on age-related changes in hippocampal pattern completion are mixed. Studies comparing hippocampal activity during episodic retrieval in older and younger adults have revealed age-related reductions in activity (*Cabeza et al., 2004*; *Dennis et al., 2008*) and age-invariant effects (*Wang et al., 2016*; *Trelle et al., 2019*). Similarly, while some have identified reduced category-level (*McDonough et al., 2014*; *Abdulrahman et al., 2017*) and event-level (*St-Laurent et al., 2014*; *Folville et al., 2020*) cortical reinstatement in older relative to younger adults, others observed age-invariant category-level reinstatement (*Wang et al., 2016*) or that age-related differences in reinstatement strength are eliminated after accounting for the strength of category representations during encoding (*Johnson et al., 2015*). Although extant studies have yielded important initial insights, the absence of trial-wise analyses relating hippocampal activity to cortical reinstatement, or relating each of these neural measures to memory behaviour, prevents clear conclusions regarding the degree to which hippocampal pattern completion processes are impacted with age. Aging may affect one or both of these neural measures, and/or may disrupt the predicted relationships between these neural variables and behaviour (e.g., *Gordon et al., 2014*). The first aim of the present study is to quantify trial-wise fluctuations in hippocampal activity and cortical reinstatement in older adults, and examine how these measures relate to one another, as well as how these measures relate to episodic remembering of trial-unique associative content.

Critically, in addition to varying within individuals, the degree to which pattern completion processes are disrupted among older adults may vary across individuals. Indeed, age-related memory decline is characterized by striking heterogeneity, with some individuals performing as well as younger adults and others demonstrating marked impairment (*de Chastelaine et al., 2016*; *Henson et al., 2016*; see *Nyberg et al., 2012* for review). Identifying the neural factors driving this variability is a clear emerging aim of cognitive aging research (*Nyberg et al., 2012*; *Cabeza et al., 2018*). However, due to modest sample sizes, extant studies typically lack sufficient power to examine individual differences in retrieval mechanisms among older adults (*Dennis et al., 2008*; *McDonough et al., 2014*; *St-Laurent et al., 2014*; *Johnson et al., 2015*; *Wang et al., 2016*; *Abdulrahman et al., 2017*; *Trelle et al., 2019*; *Folville et al., 2020*). Moreover, while recent work examining variability in hippocampal function has demonstrated relationships between hippocampal retrieval activity and associative memory performance in older adults (*de Chastelaine et al., 2016*; *Carr et al., 2017*), the direction of this relationship differed across studies; to date, the relationship between individual differences in cortical reinstatement and memory performance remains unexplored. As such, the second aim of the present study is to examine whether hippocampal and cortical indices of pattern completion vary with age, and to assess the degree to which these measures explain individual differences in episodic memory performance — both as a function of age and independent of age.

To address these two aims, a large sample (N = 100) of cognitively normal older participants (60–82 years) from the Stanford Aging and Memory Study (SAMS; *Table 1*; Materials and methods) performed an associative memory task (*Figure 1*) concurrent with high-resolution fMRI. Participants intentionally studied trial-unique word-picture pairs (concrete nouns paired with famous faces and famous places), and then had their memory for the word-picture associations probed. During retrieval scans, participants viewed a studied or novel word on each trial and indicated whether they

**Table 1.** Demographics and neuropsychological test performance.

| Measure | Mean (SD) | Range |
|---|---|---|
| Gender | 61 F; 39 M | – |
| Age (yrs) | 67.96 (5.47) | 60–82 |
| Education (yrs) | 16.84 (1.94) | 12–20 |
| MMSE | 29.10 (.90) | 26–30 |
| CDR | 0 | – |
| Logical Memory Delayed Recall (/50) | 32.04 (6.16) | 18–44 |
| HVLT-R Delayed Recall (/12) | 10.49 (1.68) | 5–12 |
| BVMT-R Delayed Recall (/12) | 9.80 (2.16) | 5–12 |
| Old/New $d'$ | 2.26 (0.68) | 0.86–4.78 |
| Associative $d'$ | 1.64 (0.73) | −0.27–3.92 |
| Exemplar-Specific Recall (proportion correct, post-scan) | 0.29 (0.19) | 0.00–0.84 |

BVMT-R = Brief Visuospatial Memory Test-Revised; CDR = Clinical Dementia Rating; HVLT-R = Hopkins Verbal Learning Test-Revised; MMSE = Mini Mental State Examination. See **Supplementary file 1** for summary of full neuropsychological test battery scores, and **Supplementary file 1** for a summary of retrieval reaction time data and trial counts by memory outcome.

The online version of this article includes the following source data for Table 1:

**Source data 1.** Demographic information and behavioural data presented in *Table 1*.

(a) recollected the associate paired with the word, responding 'face' or 'place' accordingly (providing an index of associative memory), (b) recognized the word as 'old' but were unable to recall the associate (providing an index of item memory — putatively reflecting familiarity, non-criterial recollection, or a mix of the two), or (c) thought the word was 'new'. Following scanning, participants were shown the studied words again and asked to recall the specific associate paired with each word, this time explicitly providing details of the specific image (providing an index of exemplar-specific recall).

To measure pattern completion during retrieval, we used univariate and multivariate analyses focused on a priori regions of interest (ROIs; *Figure 2*). To measure hippocampal function, our primary analyses examined univariate activity in the whole hippocampus bilaterally. In addition, we measured activity in three subfields within the body of the hippocampus — dentate gyrus/CA3 (DG/CA3), CA1, and subiculum (SUB) — given prior work suggesting that aging may differentially affect individual hippocampal subfields (*Yassa et al., 2011*; *Carr et al., 2017*; *Reagh et al., 2018*) and models predicting differential subfield involvement in pattern completion, including a key role for subfield CA3 (*Nakazawa et al., 2002*; *Grande et al., 2019*). To measure cortical reinstatement, we focused on two cortical regions — ventral temporal cortex (VTC) and angular gyrus (ANG) — which we predicted would support content-rich representations during memory retrieval based on prior evidence in healthy younger adults. In particular, while VTC has traditionally been associated with content coding during memory encoding and retrieval (*Nyberg et al., 2000*; *Wheeler et al., 2000*; *Polyn et al., 2005*; *Johnson and Rugg, 2007*; *Staresina et al., 2012*; *Ritchey et al., 2013*; *Kuhl and Chun, 2014*; *Gordon et al., 2014*; *Gagnon et al., 2019*), more recent studies have also demonstrated evidence for cortical reinstatement of both category and stimulus/event-specific features in ANG during episodic retrieval, and suggest that these representations may be differentially related to memory-guided behaviour (*Kuhl et al., 2013*; *Kuhl and Chun, 2014*; *Favila et al., 2018*; *Lee et al., 2019*). Category-level reinstatement (i.e., face/place) was quantified via pattern classification and event-specific reinstatement (e.g., Queen Elizabeth, Golden Gate Bridge) was quantified using encoding-retrieval pattern similarity.

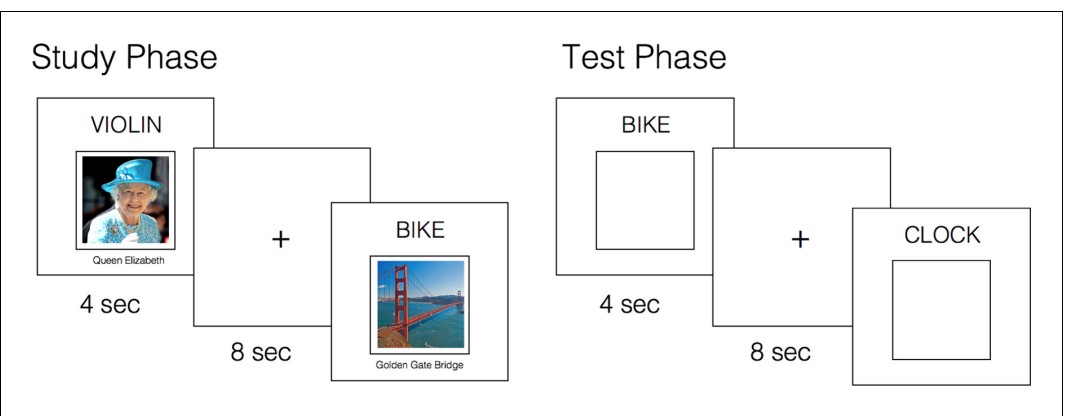

**Figure 1.** Experimental paradigm. Concurrent with fMRI, participants intentionally encoded word-picture pairs and completed an associative cued recall test. At test, they were presented with studied words intermixed with novel words, and instructed to recall the associate paired with each word, if old. Participants responded 'Face' or 'Place' if they could recollect the associated image; 'Old' if they recognized the word but could not recollect the associate; 'New' if they believed the word was novel. A post-scan cued recall test (not shown, visually identical to the 'Test Phase') further probed memory for the specific associate paired with each studied word (see Materials and methods).

## Results

### Behavioural results

We assessed performance on the associative cued recall task using three measures: 1) old/new $d'$ — discrimination between studied and novel words during the in-scan memory test, irrespective of memory for the associate; 2) associative $d'$ — correctly remembering the category of associated images encoded with studied words, relative to falsely indicating an associative category to novel words; and 3) post-scan exemplar-specific associative recall — proportion correct recall of the specific exemplars associated with studied words. Performance on all three measures declined with age (old/new $d'$: $\beta = -0.35$, $p < 0.001$; associative $d'$: $\beta = -0.30$, $p < 0.005$, **Figure 3a**; post-scan exemplar-specific recall: $\beta = -0.34$, $p < 0.001$, **Figure 3b**), but did not vary by sex ($\beta s = -0.10, -0.33, -0.23$; $ps \geq 0.10$) or years of education ($\beta = -0.03, -0.02, -0.07$; $ps > 0.47$). Associative $d'$ was higher for word-face pairs than word-place pairs ($t(99) = 5.37$, $p < 10^{-7}$). Critically, despite this decline in performance with age, we also observed considerable variability in performance across individuals in each measure (**Figure 3** and **Table 1**).

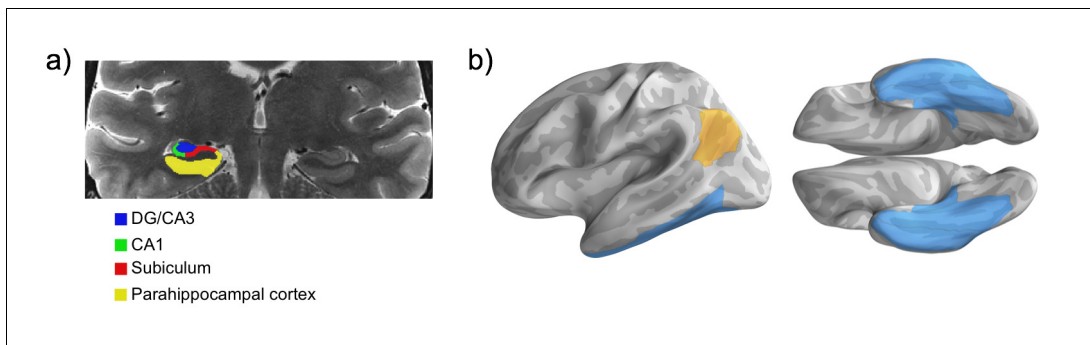

**Figure 2.** Regions of interest. (a) Sample MTL subfield demarcations. The whole hippocampus ROI reflects the summation of all subfields (delineated only in the hippocampal body, shown), as well as the hippocampal head and tail (not pictured). (b) Parahippocampal cortex combined with fusiform gyrus and inferior temporal cortex forms the ventral temporal cortex ROI. Ventral temporal cortex (blue) and angular gyrus (gold) masks projected on the fsaverage surface.

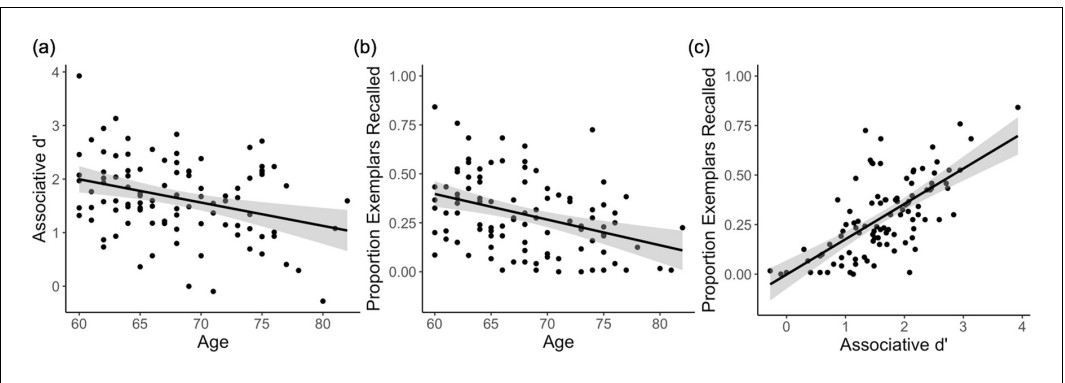

**Figure 3.** Associative memory behavioural results. (**a**) In-scanner associative *d'* and (**b**) post-scan exemplar-specific associative recall decline with age. (**c**) Associative *d'* is strongly correlated with post-scan exemplar-specific associative recall, controlling for the effect of age. Each data point represents a participant; plots show linear model predictions (black line) and 95% confidence intervals (shaded area).

The online version of this article includes the following source data for figure 3:

**Source data 1.** Demographic information and behavioural data depicted in *Figure 3a–c*.

Individual-differences and trial-wise analyses revealed that post-scan associative recall tracked in-scanner associative memory. First, individuals who demonstrated higher associative memory during scanning showed superior recall of the specific exemplars on the post-scan test (controlling for age; $\beta = 0.62$, $p < 10^{-12}$; *Figure 3c*). Second, trial-wise analysis revealed that making an in-scan associative hit was a significant predictor of successful post-scan exemplar recall ($\chi^2(1) = 159.68$, $p < 10^{-36}$). These findings suggest that post-scan exemplar-specific retrieval — while quantitatively lower due to the longer retention interval, change of context, and interference effects — is a good approximation of recall of the specific exemplar during scanning (relative to simply recalling more general category information).

## fMRI encoding classifier accuracy

Following prior work (e.g., *Kuhl et al., 2013*; *Kuhl and Chun, 2014*; *Favila et al., 2018*; *Lee et al., 2019*), cortical reinstatement analyses focused on two a priori ROIs: VTC and ANG. To confirm that activity patterns during word-face and word-place encoding trials were discriminable for each participant in each ROI, we trained and tested a classifier on the encoding data using leave-one-run-out-n-fold cross validation. On average, encoding classifier accuracy was well above chance (50%) using patterns in VTC (M = 98.4%, $p < 0.001$) and ANG (90.0%, $p < 0.001$), with classifier accuracy significantly greater in VTC than ANG ($t(99) = 12.86$, $p < 10^{-16}$). Classification was above chance in all 100 participants (minimum accuracy of 82.5% ($p < 0.001$) in VTC and 68.0% ($p < 0.005$) in ANG), and did not vary significantly as a function of age (VTC: $\beta = -0.13$, $p = 0.133$; ANG: $\beta = -0.06$, $p = 0.544$). To account for variance in encoding classifier strength (quantified using log odds of the classifier's probability estimate) on estimates of category-level reinstatement strength during memory retrieval (trial-wise: VTC: $\chi^2(1) = 13.96$, $p < 0.001$; ANG: $\chi^2(1) = 30.16$, $p < 10^{-8}$; individual differences: VTC: $\beta = 0.45$, $p < 10^{-5}$; ANG: $\beta = 0.62$, $p < 10^{-11}$; see *Figure 5—figure supplement 3*), we controlled for encoding classifier strength in all subsequent models in which category-level reinstatement strength was related to behavioural variables (memory accuracy, RT), as well as in models in which reinstatement strength was the dependent variable (see Materials and methods – Statistical Analysis and *Supplementary file 1* for details).

## Memory behaviour scales with trial-wise category-level reinstatement

We quantified reinstatement of relevant face or scene features (i.e., category-level reinstatement) in VTC and ANG using subject-specific classifiers trained on all encoding phase runs for an individual (training set was balanced for category), and tested for cortical reinstatement in the independent retrieval phase data; significance was assessed using permutation testing (see Materials and methods – MVPA for further details). Classifier accuracy (*Figure 4a*) was above chance

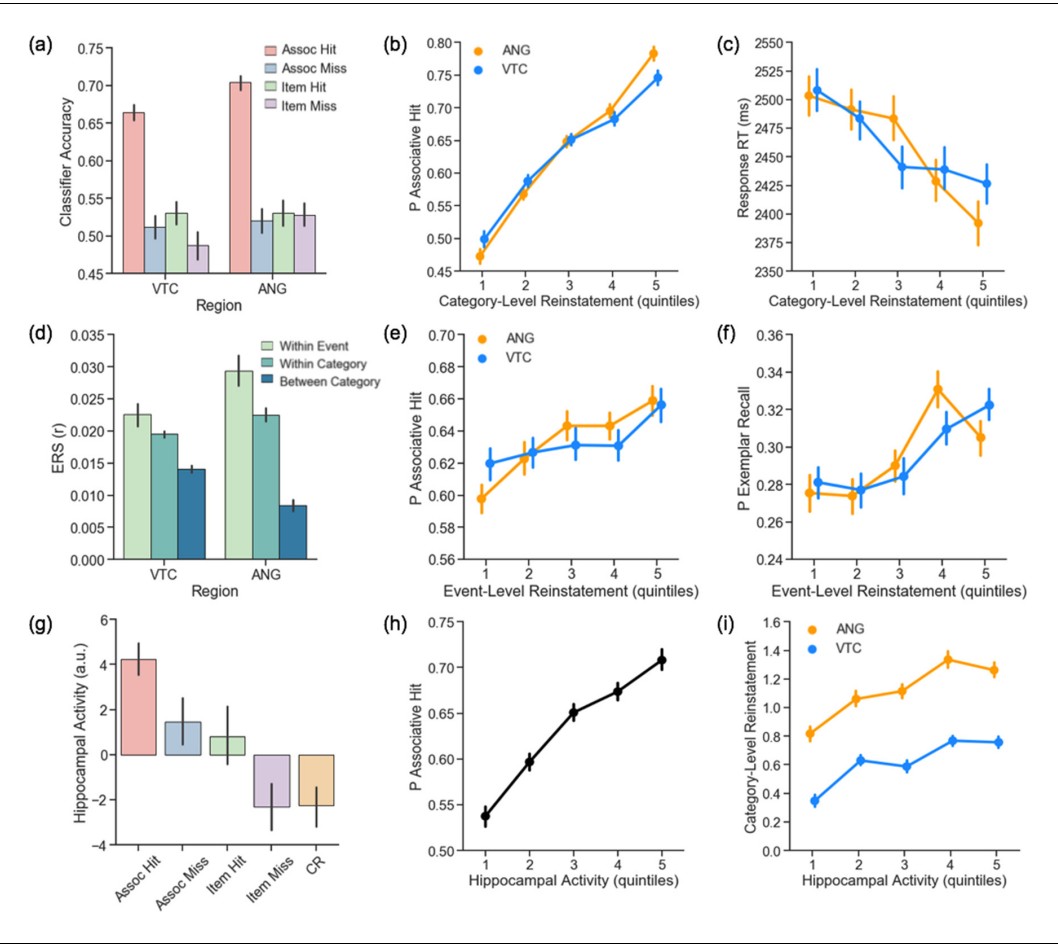

**Figure 4.** Cortical and hippocampal metrics of pattern completion during retrieval. (**a**) Classifier accuracy is above chance in VTC and ANG during successful, but not unsuccessful, associative retrieval. (**b**) Trial-wise category-level reinstatement strength (logits) in VTC and ANG is related to an increased probability of an associative hit and (**c**) faster decision RT on associative hit trials. (**d**) Event-level reinstatement (within-event ERS > within-category ERS) is observed during associative hits in VTC and ANG. (**e**) Trial-wise event-level reinstatement (within-event ERS) significantly varies with the probability of an associative hit and (**f**) exemplar-specific hit. (**g**) Hippocampal activity shows a graded response across retrieval conditions. (**h**) Trial-wise hippocampal activity is related to an increased probability of an associative hit and (**i**) greater category-level reinstatement strength (logits) in VTC and ANG. For visualization, data for each participant are binned into quintiles based on category-level reinstatement strength (**b, c**), event-level reinstatement strength (**e,f**) and hippocampal activity (**h,i**). Statistics were conducted on trial-wise data, z-scored within participant. Error bars represent standard error of the mean. VTC = ventral temporal cortex; ANG = angular gyrus; RT = reaction time; ERS = Encoding Retrieval Similarity.

The online version of this article includes the following source data and figure supplement(s) for figure 4:

**Source data 1.** Classifier accuracy in VTC and ANG by trial type, depicted in *Figure 4a*.

**Source data 2.** Trial-wise cortical reinstatement (logits), hippocampal activity, and behavioural data used to generate *Figure 4b–c,h–i* and *Figure 4—figure supplements 3*, *5* and *6*.

**Source data 3.** Encoding-retrieval similarity in VTC and ANG by trial type, depicted in *Figure 4d* and *Figure 4—figure supplement 2a*.

**Source data 4.** Trial-wise encoding-retrieval similarity, hippocampal activity, and behavioural data used to generate *Figure 4e–f* and *Figure 4—figure supplements 2b–c* and *4*.

**Source data 5.** Hippocampal activity by trial type, depicted in *Figure 4g*.

**Figure supplement 1.** Time course of cortical reinstatement during associative hits.

**Figure supplement 2.** Replication of category-level reinstatement effects computed via encoding-retrieval similarity.

**Figure supplement 3.** In-scanner pattern completion metrics are related to post-scan exemplar-specific recall.

**Figure supplement 4.** Trial-wise hippocampal activity is related to within-event ERS in VTC.

*Figure 4 continued on next page*

*Figure 4 continued*

**Figure supplement 5.** Hippocampal subfield activity during associative retrieval.
**Figure supplement 6.** Hippocampal head and tail activity during associative retrieval.

(50%) during associative hits in VTC (M = 68.3%, $p$ < 0.005) and ANG (M = 72.3%, $p$ < 0.001), but did not exceed chance when associative retrieval failed, including on associative miss trials (VTC: 49.8%, $p$ = 0.57; ANG: 50.4%, $p$ = 0.49), item hit trials (VTC: 53.5%, $p$ = 0.29; ANG: 53.3%, $p$ = 0.31), and item miss trials (VTC: 47.1%, $p$ = 0.68; ANG: 51.6%, $p$ = 0.41; see Materials and methods for trial type definitions). Classifier accuracy during associative hits was greater in ANG relative to VTC ($t$(99) = 3.96, $p$ < 0.001). In VTC, classifier accuracy during associative hits was stronger on place trials (M = 71.5%) relative to face trials (M = 65.1%, $t$(99) = 5.25, $p$ < $10^{-7}$), whereas in ANG the strength of reinstatement did not significantly vary by stimulus category (place: M = 73.3%; face: M = 71.3%, $t$(99) = 1.69, $p$ = 0.094). To control for possible effects of stimulus category on the results, category is included as a regressor in all linear and logistic mixed effects models, and interactions between category and primary variables of interest are examined and reported in *Supplementary file 1*). Analyses of the time course of cortical reinstatement during associative hits revealed significant category-level reinstatement effects emerging ~4–6 s post-stimulus onset (*Figure 4—figure supplement 1*). Analogous category-level reinstatement effects were observed using a pattern similarity approach (i.e., encoding-retrieval similarity (ERS); see *Figure 4—figure supplement 2*).

Evidence for reinstatement during successful, but not unsuccessful, associative retrieval is consistent with theories that posit that reinstatement of event features (here, face or scene features) supports accurate memory-based decisions (here, associate category judgments). More directly supporting this hypothesis, generalized logistic and linear mixed effects models (see *Supplementary file 1* for full list of model parameters) revealed that greater trial-wise category-level cortical reinstatement in VTC and ANG — quantified using log odds of the classifier's probability estimate — was related to (a) an increased probability of an associative hit (VTC: $\chi^2$(1) = 102.18, $p$ < $10^{-24}$; ANG: $\chi^2$(1) = 133.25, $p$ < $10^{-31}$; *Figure 4b*), (b) an increased probability of post-scan exemplar-specific recall (VTC: $\chi^2$(1) = 62.85, $p$ < $10^{-15}$; ANG: $\chi^2$(1) = 89.02, $p$ < $10^{-21}$; *Figure 4—figure supplement 3*), and (c) faster retrieval decision RTs on associative hit trials (VTC: $\chi^2$(1) = 30.08, $p$ < $10^{-8}$; ANG: $\chi^2$(1) = 21.73, $p$ < $10^{-6}$; *Figure 4c*). We also found that age moderated the relationship between category-level reinstatement strength in VTC and behaviour, such that older individuals exhibited a weaker relationship between reinstatement strength in VTC and (a) associative retrieval success ($\chi^2$(1) = 7.12, $p$ < 0.01) and (b) retrieval decision RT on associative hit trials ($\chi^2$(1) = 3.91, $p$ < 0.05). This interaction was marginally significant in ANG with respect to associative retrieval ($\chi^2$(1) = 3.57, $p$ = 0.059), but not decision RT ($\chi^2$(1) = 0.16, $p$ = 0.685). Together, these data provide novel evidence that the strength of category-level reinstatement in VTC and ANG is linked to memory behaviour in cognitively normal older adults (see *Figure 4—figure supplement 2* for analogous ERS findings), and also suggest that older age negatively impacts the translation of cortical evidence to memory behaviour.

## Memory behaviour scales with trial-wise event-level reinstatement

We next used encoding-retrieval similarity (ERS) to quantify trial-unique, event-specific reinstatement of encoding patterns, comparing the similarity of an event's encoding and retrieval patterns (within-event ERS) to similarity of encoding patterns from other events from the same category (within-category ERS). Evidence for event-level reinstatement was present in both VTC ($t$(99) = 2.26, $p$ < 0.05) and ANG ($t$(99) = 3.54, $p$ < 0.001) during associative hits (*Figure 4d*). Moreover, the strength of trial-wise event-level reinstatement — controlling for within-category ERS (see *Supplementary file 1* for full list of model parameters)— was related to (a) an increased probability of an associative hit (VTC: $\chi^2$(1) = 1.78, $p$ = 0.183; ANG: $\chi^2$(1) = 7.50, $p$ = 0.006; *Figure 4e*) and (b) an increased probability of post-scan exemplar-specific recall (VTC: $\chi^2$(1) = 5.35, $p$ < 0.05; ANG: $\chi^2$(1) = 7.27, $p$ = 0.006; *Figure 4f*), but not with decision RT on associative hit trials (VTC: $p$ = 0.845; ANG: $p$ = 0.231). These relationships were not significantly moderated by age (all $p$ > 0.254). These results

demonstrate a relationship between trial-unique, event-specific cortical reinstatement and associative retrieval in older adults.

## Behaviour and reinstatement scale with trial-wise hippocampal retrieval activity

Successful associative retrieval, ostensibly driven by pattern completion, was accompanied by greater hippocampal activity (*Figure 4g*) relative to associative misses ($t(75) = 4.90$, $p < 10^{-6}$), item only hits ($t(59) = 3.87$, $p < 0.001$), item misses ($t(83) = 8.86$, $p < 10^{-13}$), and correct rejections ($t(99) = 11.28$, $p < 10^{-16}$). Relative to item misses, hippocampal activity was greater during associative misses ($t(68) = 4.0$, $p < 0.001$) and item only hits ($t(51) = 5.37$, $p < 10^{-6}$); activity did not differ between associative misses and item hits ($t < 1$) or between item misses and correct rejections ($t < 1$). Moreover, generalized logistic and linear mixed effects models revealed that greater trial-wise hippocampal activity was related to (a) an increased probability of an associative hit ($\chi^2(1) = 63.45$, $p < 10^{-15}$; *Figure 4h*), (b) an increased probability of post-scan exemplar-specific recall ($\chi^2(1) = 59.02$, $p < 10^{-14}$; *Figure 4—figure supplement 3*), but (c) not faster associative hit RTs ($\chi^2(1) = 2.08$, $p = 0.149$). These relationships were not moderated by age (associative hit: $p = 0.616$; exemplar-specific recall: $p = 0.713$). Thus, the probability of successful pattern-completion-dependent associative retrieval increased with hippocampal activity. This relationship was significant across hippocampal subfields, but greatest in DG/CA3 (see *Figure 4—figure supplements 5–6* for subfield findings).

Cortical reinstatement is thought to depend on hippocampal pattern completion triggered by retrieval cues (*Marr, 1971*; *McClelland et al., 1995*; *Tanaka et al., 2014*; *Staresina et al., 2019*). Consistent with this possibility, the magnitude of trial-wise hippocampal retrieval activity significantly varied with the strength of category-level cortical reinstatement across all retrieval attempts (VTC: $\chi^2(1) = 43.36$, $p < 10^{-11}$; ANG: $\chi^2(1) = 35.31$, $p < 10^{-9}$; *Figure 4i*) and when restricting analyses only to associative hit trials (VTC: $\chi^2(1) = 5.77$, $p < 0.05$; ANG: $\chi^2(1) = 9.48$, $p < 0.005$). Similarly, hippocampal activity significantly varied with within-event ERS (controlling for within-category ERS) in VTC (all trials: $\chi^2(1) = 4.55$, $p < 0.05$; associative hit only: $\chi^2(1) = 3.73$, $p = 0.054$; see *Figure 4—figure supplement 4*); this relationship did not reach significance in ANG (all trials: $p = 0.328$; associative hit only: $p = 0.289$). The relationship between hippocampal activity and reinstatement strength was not moderated by age (category-level reinstatement VTC: $p = 0.777$; ANG: $p = 0.773$; event-level reinstatement VTC: $p = 0.493$). Collectively, these results constitute novel evidence for a relationship between trial-wise hippocampal activity and cortical reinstatement in older adults. This relationship was also observed in select hippocampal subfields (see *Figure 4—figure supplements 5–6*).

## Cortical reinstatement partially mediates the effect of hippocampal activity on retrieval

Having established a relationship between associative retrieval success and (a) hippocampal activity, (b) cortical reinstatement strength in VTC, and (c) ANG, we next sought to determine whether each of these putative indices of pattern completion explain common or unique variance in associative retrieval success. Using nested comparison of logistic mixed effects models, we found that compared to a model with image category and hippocampal activity, addition of VTC category-level reinstatement strength significantly improved model fit ($\chi^2(1) = 103.48$, $p < 10^{-24}$). Addition of ANG category-level reinstatement to this model further improved model fit ($\chi^2(1) = 115.42$, $p < 10^{-27}$), and all three variables remained significant predictors in the full model (hippocampus: $b = 0.31$, $z = 8.36$, $p < 10^{-16}$; VTC: $b = 0.32$, $z = 9.36$, $p < 10^{-16}$; ANG: $b = 0.52$, $z = 14.36$, $p < 10^{-16}$). These results indicate that reinstatement strength and hippocampal activity, though related indices of pattern completion, nevertheless explain unique variance in the probability of a successful associative retrieval decision. Moreover, they indicate that measures of category-level reinstatement strength in different cortical regions are not redundant, and perhaps carry complementary information relevant for memory behaviour.

Given our prediction that the present measures of cortical reinstatement are, at least in part, a read out of hippocampal pattern completion processes, we next sought to more directly test the hypothesis that cortical reinstatement mediates the relationship between hippocampal activity and associative retrieval success. We conducted a mediation analysis separately for each cortical ROI, in

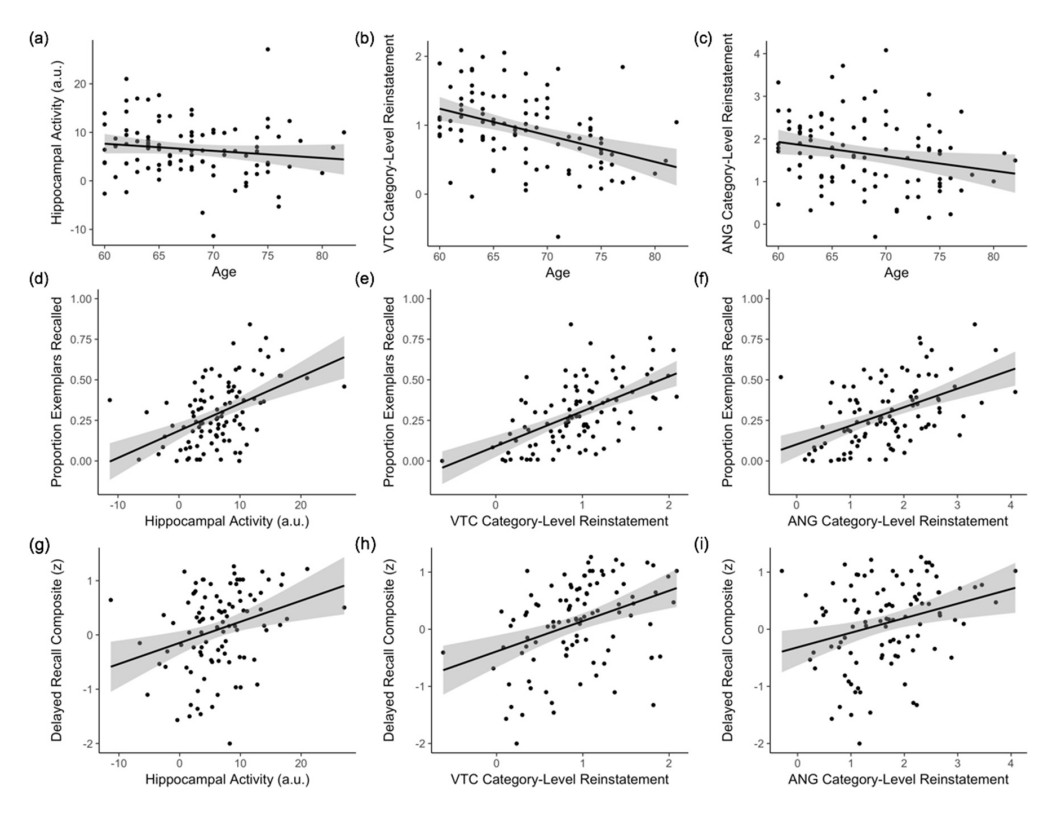

**Figure 5.** Individual differences in pattern completion assays. (**a–c**) Effects of age on hippocampal activity (associative hit – correct rejection) and category-level reinstatement strength (mean logits) in VTC and ANG during associative hits. (**d–f**) Independent of age, individual differences in hippocampal activity and category-level reinstatement strength in VTC and ANG during associative hits explain significant variance in exemplar-specific recall. (**g–i**) Independent of age, individual differences in hippocampal activity and VTC category-level reinstatement strength also explain significant variance in standardized delayed recall performance; the relation with ANG category-level reinstatement did not reach significance. Scatterplots reflect raw values for each measure. See **Figure 5—figure supplement 1** for partial plots controlling for nuisance variables. Each point represents an individual participant. Plots also show linear model predictions (black line) and 95% confidence intervals (shaded area). VTC = ventral temporal cortex; ANG = angular gyrus.

The online version of this article includes the following source data and figure supplement(s) for figure 5:

**Source data 1.** Behavioural and neural measures used in individual differences analyses, depicted in **Figure 5a–i** and **Figure 5—figure supplements 1–5**.

**Figure supplement 1.** Partial plots of individual differences in pattern completion assays.

**Figure supplement 2.** Event-Level reinstatement strength (within-event – within-category ERS) during associative hits declines with age in (**a**) VTC, but not (**b**) ANG.

**Figure supplement 3.** Relationship between Encoding Strength and Category-Level Reinstatement Strength in (**a**) VTC and (**b**) ANG.

**Figure supplement 4.** Individual differences in pattern completion assays predict associative *d'*.

**Figure supplement 5.** Pattern Completion Metrics Covary across Participants.

which the coefficient of the indirect path was computed as the product of the direct effects, *a* x *b*, and the significance of the indirect effect was calculated using bootstrap resampling (see Materials and methods – Statistics for details). Consistent with predictions, the results revealed that the relationship between hippocampal activity and the probability of an associative hit was partially mediated through category-level cortical reinstatement in VTC (indirect effect: *b* = 0.026, 95% CI = 0.016, 0.036) and ANG (indirect effect: *b* = 0.019, 95% CI = 0.006, 0.032). These findings demonstrate that the effect of retrieval-phase hippocampal activity on associative retrieval success can be explained in part through its effects on cortical reinstatement.

## Effects of age on hippocampal and cortical indices of pattern completion

Our second key aim was to understand how hippocampal pattern completion processes vary across individuals, turning first to the effects of age. For all individual-differences analyses of pattern completion, we computed mean category-level and event-level reinstatement strength in VTC and ANG during associative hits and mean hippocampal activity during successful associative hits (corrected by mean activity during correct rejections) for each participant. Each measure was adjusted by head motion, and reinstatement strength was further adjusted by encoding strength, before it was entered into regression models. Regression analyses revealed that (a) while hippocampal activity did not significantly vary with age ($\beta = -0.10$, $p = 0.35$; *Figure 5a*), there was (b) an age-related decline in category-level reinstatement strength during associative hits (i.e., mean logits; VTC: $\beta = -0.34$, $p < 0.001$; ANG: $\beta = -0.16$, $p < 0.05$; *Figure 5b–c*), and c) an age-related decline in event-level reinstatement (i.e., ERS) during associative hits in VTC ($\beta = -0.26$, $p < 0.01$; *Figure 5—figure supplement 2*), but not ANG ($\beta = -0.06$, $p = 0.55$; *Figure 5—figure supplement 2*). None of these measures varied with sex or years of education (all $ps > 0.24$).

## Neural indices of pattern completion explain individual differences in episodic memory

We next asked if the strength of neural measures of pattern completion during associative retrieval explain variance in memory performance, independent of age. Separate regression models, controlling for age, revealed that individual differences in exemplar-specific recall were related to hippocampal activity ($\beta = 0.47$, $p < 10^{-7}$; *Figure 5d*) and category-level reinstatement strength during associative hits (VTC: $\beta = 0.45$, $p < 10^{-6}$; ANG: $\beta = 0.41$, $p < 0.001$, *Figure 5e–f*; see *Figure 5—figure supplement 1* for partial plots adjusting for nuisance regressors and *Figure 5—figure supplement 4* for parallel findings with associative $d'$). In contrast, individual differences in event-level reinstatement did not explain significant variance in exemplar-specific recall (all $ps > 0.33$). Thus, individual differences in the integrity of hippocampal retrieval mechanisms and category-level cortical reinstatement contribute to variability in pattern-completion-dependent (i.e., associative) memory in older adults. To determine if these observed effects were moderated by age, we repeated analyses including an age × predictor interaction in each model. These models provided no significant evidence for an age-related moderation of the effect of hippocampal activity ($\beta = -0.15$, $p = 0.088$) or category-level reinstatement strength (VTC: $p = 0.977$; ANG: $p = 0.565$) on exemplar-specific recall. While these results suggest that the strength of the relationships between (a) hippocampal activity and (b) category-level reinstatement strength and individual differences in associative memory is age-invariant, we interpret this result with caution given the restricted age range (60–82 years) of the current sample.

To determine whether these neural variables explain unique variance in memory performance, we used hierarchical regression (see *Table 2* for model parameters). Compared to a model with age alone (adjusted $R^2 = 0.126$), adding hippocampal activity explained additional variance in exemplar specific recall (model comparison: $F(1,96) = 29.54$, $p < 10^{-7}$, adjusted $R^2 = 0.325$). Moreover, adding a single category-level reinstatement metric explained further variance in performance (model comparison: VTC: $F(1,95) = 22.75$, $p < 10^{-6}$, adjusted $R^2 = 0.438$; ANG: $F(1,95) = 8.25$, $p < 0.01$, adjusted $R^2 = 0.365$). However, when VTC and ANG were both included in the same model, category-level reinstatement strength in ANG was no longer a significant predictor ($p = 0.412$). Analogous findings were observed with associative $d'$ as the dependent variable (see *Supplementary file 1* for model parameters). Thus, in older adults, individual differences in hippocampal activity and cortical reinstatement strength provide complementary information, over and above age, in explaining individual differences in associative memory, whereas indices of category-level reinstatement strength explain shared variance.

## Independent measures of memory scale with individual differences in pattern completion

Finally, we examined whether our task-based fMRI measures of pattern completion — hippocampal activity and cortical reinstatement — explain individual differences in an independent measure of episodic memory, using a delayed recall composite score collected in a separate neuropsychological

**Table 2.** Summary of regression analysis predicting post-test exemplar-specific recall.

| | Variable | β | SE | p | Adjusted R² |
|---|---|---|---|---|---|
| Step 1 | Age | −0.366 | 0.094 | 0.001*** | 0.126 |
| Step 2 | Age | −0.317 | 0.083 | 0.001*** | 0.325 |
| | Hippocampal Activity[a] | 0.472 | 0.087 | 0.001*** | |
| Step 3a | Age | −0.184 | 0.080 | 0.023* | 0.449 |
| | Hippocampal Activity[a] | 0.388 | 0.080 | 0.001*** | |
| | VTC Reinstatement[ab] | 0.428 | 0.089 | 0.001*** | |
| Step 3b | Age | −0.281 | 0.082 | 0.001*** | |
| | Hippocampal Activity[a] | 0.407 | 0.088 | 0.001**** | 0.365 |
| | ANG Reinstatement[ab] | 0.289 | 0.108 | 0.009** | |
| Step 4 | Age | −0.184 | 0.080 | 0.023*** | 0.448 |
| | Hippocampal Activity[a] | 0.374 | 0.082 | 0.001*** | |
| | VTC Reinstatement[ab] | 0.391 | 0.100 | 0.001*** | |
| | ANG Reinstatement[ab] | 0.093 | 0.113 | 0.412 | |
| Step 5 | Age | −0.137 | 0.079 | 0.087~ | 0.485 |
| | Hippocampal Activity[a] | 0.335 | 0.080 | 0.001**** | |
| | VTC Reinstatement[ab] | 0.377 | 0.089 | 0.001**** | |
| | Delayed Recall | 0.299 | 0.110 | 0.008** | |

*Note.* [a] = adjusted by motion; [b] = adjusted by encoding strength (mean logits across leave-one-run-out-n-fold cross validation); Reinstatement = category level reinstatement (mean logits across associative hits); SE = standard error; VTC = ventral temporal cortex; ANG = angular gyrus; ~p < 0.1, *p<0.05, **p<0.01, ***p<0.001 ****p<10$^{-5}$.

testing session (see Materials and methods). Controlling for age and sex, hippocampal activity ($β = 0.19$, $p < 0.01$; *Figure 5g*) and VTC category-level reinstatement strength ($β = 0.21$, $p <0.01$; *Figure 5h*) were significant predictors of delayed recall score; the relationship with ANG category-level reinstatement strength did not reach significance ($β = 0.14$, $p = 0.11$; *Figure 5i*; see *Figure 5—figure supplement 1* for partial plots). Further, as for exemplar-specific recall, we found that hippocampal activity and VTC category-level reinstatement strength explained unique variance in delayed recall performance (hippocampus: $β = 0.16$, $p < 0.05$; VTC: $β = 0.20$, $p < 0.05$, adjusted $R^2 = 0.231$).

Given the observed relationships between this standardized neuropsychological measure and the present indices of pattern completion, we asked whether delayed recall score alone could account for the observed relationship between the neural measures and exemplar-specific recall. When delayed recall score was added to the full model (see *Table 2*, Step 5), this measure explained additional variance in exemplar-specific recall (model comparison: $F(1,94) = 7.45$, $p < 0.01$, adjusted $R^2 = 0.485$), but hippocampal activity and VTC category-level reinstatement strength remained significant predictors (hippocampus: $β = 0.335$, $p < 10^{-5}$; VTC reinstatement: $β = 0.377$, $p < 10^{-5}$). Together, these results support the hypothesis that individual differences in the integrity of pattern completion processes, indexed by univariate and pattern-based task-related fMRI metrics, explain variance in memory performance across established hippocampal-dependent assays of episodic memory, and do so in a manner that is not captured by simple standardized neuropsychological tests.

## Discussion

Using univariate and multivariate fMRI, the current investigation characterizes the integrity of hippocampal pattern completion during associative retrieval in a large cohort of putatively healthy older adults. We provide novel evidence for unique contributions of hippocampal and cortical indices of pattern completion to a) trial-by-trial differences in episodic remembering in older adults, as well as b) age-related and age-independent individual differences in episodic memory performance. Taken together, these results provide novel insights into the neural mechanisms supporting episodic memory, as well as those driving variability in remembering across older adults.

The present analyses of trial-level brain-behaviour relationships significantly build on work in younger adults (*Gordon et al., 2014*; *Gagnon et al., 2019*), demonstrating that trial-wise relationships (a) between hippocampal activity and cortical reinstatement and (b) between each of these neural measures and memory behaviour are present later in the lifespan. While directionality is difficult to establish with fMRI, these results are consistent with models of episodic retrieval wherein hippocampal pattern completion, triggered by partial cues, drives reinstatement of event representations in the cortex, which supports episodic remembering and memory-guided decision making (*Marr, 1971*; *McClelland et al., 1995*). Further bolstering this interpretation, we demonstrate that (a) category-level cortical reinstatement partially mediates the relationship between hippocampal activity and associative retrieval success, and (b) the relationship between hippocampal activity and associative retrieval success was qualitatively strongest in DG/CA3, consistent with a key role of CA3 in initiating pattern-completion dependent retrieval (*Marr, 1971*; *McClelland et al., 1995*; *Nakazawa et al., 2002*; *Tanaka et al., 2014*; *Staresina et al., 2019*). Moreover, the present results provide novel evidence for stability in the trial-wise relationship between hippocampal activity and (a) cortical reinstatement and (b) associative retrieval success, as neither relationship varied as a function of age.

Consistent with the observed trial-level relationship between hippocampal activity and associative retrieval success, we also demonstrate a positive relationship between the magnitude of hippocampal activity during associative hits and associative memory performance. Our findings complement and build on prior work (*de Chastelaine et al., 2016*), as we demonstrate that this effect was observed across hippocampal subfields, including DG/CA3, and did not vary significantly as a function of age. These results are compatible with proposals that the relationship between hippocampal 'recollection success' effects and memory performance remains stable across the lifespan (*de Chastelaine et al., 2016*), as well as more broadly with proposals that preservation of hippocampal function is important for the maintenance of episodic memory in older adults over time (*Persson et al., 2012*; *Pudas et al., 2013*). We note, however, that a negative relationship between hippocampal retrieval activity and memory performance has also been observed in older adults (e. g., *Carr et al., 2017*; *Reagh et al., 2018*). Differences across studies may be related to (a) the paradigms and/or contrasts employed (e.g., associative recollection vs. lure discrimination), (b) image resolution (e.g., individual subfields vs. the whole hippocampus), or (c) the make-up of the study population (e.g., cognitively normal or cognitively impaired; *Dickerson and Sperling, 2008*). Additional well-powered studies of hippocampal retrieval dynamics in older adults are needed to assess the degree to which these variables alter the relationship between hippocampal activity and memory behaviour.

The present results also provide novel insights into the basis of mnemonic decisions in older adults. Specifically, we demonstrate that trial-wise indices of reinstatement strength — indexed using classifier-derived evidence and encoding-retrieval pattern similarity — were tightly linked to memory behaviour, including response accuracy and speed. This finding suggests that retrieval was not 'all or none', but likely graded (*Mickes et al., 2009*; *Kuhl et al., 2011*; *Harlow and Yonelinas, 2016*). Indeed, while participants were instructed during scanning to recollect the specific associate, correct category judgments (agnostic to correct exemplar-specific recall) could nonetheless be supported by retrieval of generic category information (e.g., a place), prototypical details (e.g., a bridge), specific exemplar details (e.g., the Golden Gate Bridge), or even retrieval of erroneous, but category consistent details (e.g., Niagara Falls). The category-level reinstatement effects observed here likely reflect some combination of these retrieval outcomes, as suggested by the strong correlation between post-scan exemplar-specific recall and within-scan associative *d'*, along with the observation that the proportion of specific exemplars recalled post-scan was generally lower than correct

categorical judgements during scanning (though the former undoubtedly declined due to the longer retention interval and interference effects).

Beyond the strength of reinstatement, the present results cannot adjudicate the nature of the details recalled. For example, both category- and exemplar-specific associative hits could be supported by retrieval of semantic details (e.g., the Golden Gate Bridge), perceptual details (e.g., the bridge was red), or some combination (e.g., vividly recalling the image of the Golden Gate Bridge). One possibility, though speculative, is that VTC and ANG support representations of distinct types of event features (e.g., perceptual features in VTC and semantic and/or multimodal features in ANG). This possibility is in line with existing evidence (e.g., *Bonnici et al., 2016*; *Favila et al., 2018*) and also with the present observation that reinstatement strength in VTC and ANG made complementary contributions to retrieval success. Regardless of the precise nature of the details recalled, we demonstrate that, as in younger adults (*Kuhl et al., 2011*; *Kuhl et al., 2013*; *Gordon et al., 2014*), recovery of stronger mnemonic evidence was associated with greater accuracy and faster responses, and this was true for representations supported by VTC and ANG alike. This relationship may reflect reduced demands on post-retrieval monitoring and selection processes and/or greater confidence in the face of stronger mnemonic evidence. Interestingly, the strength of the trial-level relationship between VTC reinstatement strength and behaviour weakened with increased age. This could be related to age-related changes in decision criteria, retrieval monitoring ability, response strategies, or some combination of these factors. Future work is needed to explore the specific neurocognitive basis of this intriguing effect, which likely involves interactions between the medial temporal lobe and frontoparietal regions (*Waskom et al., 2014*; *Gagnon et al., 2019*).

Although we observed robust group-level cortical reinstatement effects during associative hits, category-level reinstatement strength declined with age, and individual differences in category-level reinstatement strength explained significant variance in episodic memory. Importantly, the effect of age on reinstatement strength, and the relationship between reinstatement strength and memory performance, was observed after accounting for variance in encoding classifier performance, a putative assay of cortical differentiation (i.e., the ability to establish distinct neural patterns associated with different visual stimulus categories) during memory encoding. Prior work has demonstrated reductions in cortical differentiation in older relative to young adults (*Voss et al., 2008*; *Carp et al., 2011*; *Park et al., 2012*; *Koen et al., 2019*; *Trelle et al., 2019*), and evidence from both older and young adults suggests that cortical differentiation at encoding can impact reinstatement strength (*Gordon et al., 2014*; *Johnson et al., 2015*) and memory performance (*Koen et al., 2019*). Indeed, we found that encoding classifier strength was a strong predictor of category-level reinstatement strength in the present sample. Critically, by controlling for encoding strength in the current analyses, the present results indicate that the observed variance in reinstatement strength, and its relation to memory performance, does not simply reflect downstream effects of cortical differentiation. Instead, variance in reinstatement strength likely also provides information about the precision with which event representations are retrieved in older adults. These data therefore provide neuroimaging evidence in support of existing proposals that age-related episodic memory decline is driven, in part, by a loss of specificity or precision in mnemonic representations, a possibility that has been well-supported by behavioural evidence (*Luo and Craik, 2009*; *Trelle et al., 2017*; *Korkki et al., 2020*).

Interestingly, while cortical reinstatement is a putative read-out of pattern completion, and therefore relies critically on the hippocampus – a possibility supported by the present data – the hippocampal and cortical measures of pattern completion defined here explained unique variance in memory performance, both at the trial level and across individuals. Indeed, these measures together explained nearly three times as much variance in exemplar-specific associative recall as age alone. One possibility is that hippocampal activity and cortical reinstatement strength index distinct aspects of recollection: retrieval success vs. retrieval precision, respectively (e.g., *Harlow and Yonelinas, 2016*; *Richter et al., 2016*). That is, whereas increases in hippocampal activity may signal recollection of some event details, this signal alone may not indicate the fidelity or precision with which the event is recollected. Conversely, reinstatement strength may provide more information about the contents of recollection, including the specificity or precision of mnemonic representations (e.g., recall of generic as opposed to exemplar-specific details), and perhaps even the nature of the details recollected (i.e., perceptual vs semantic). An alternative, but not mutually exclusive possibility, is that representations reinstated in cortex may be differentially affected by top-down goal states, post-

retrieval monitoring, selection and/or decision processes (*Kuhl et al., 2013*; *Favila et al., 2018*), which may contribute unique variance in memory performance beyond that explained by hippocampal-initiated event replay. Future work is needed to examine whether the unique variance explained by cortical reinstatement relates to frontoparietal control and decision processes in older adults.

Indeed, it is important to note that variability in episodic remembering, and indeed variability in the strength of the present pattern completion metrics, is likely influenced by a number of variables, only some of which are measured here. For example, aging may affect other processes at retrieval, including elaboration of retrieval cues (*Morcom and Rugg, 2004*) and post-retrieval monitoring and selection (*McDonough et al., 2013*; *Trelle et al., 2019*), as well as factors at encoding, including the differentiation of stimulus representations (*Voss et al., 2008*; *Carp et al., 2011*; *Park et al., 2012*; *Koen et al., 2019*; *Trelle et al., 2019*), goal-directed or sustained attention (*Hultsch et al., 2002*; *Geerligs et al., 2014*), and elaborative or 'strategic' encoding processes (*Luo et al., 2007*; *Trelle et al., 2015*). These variables could vary both within individuals (i.e., across trials), as well as between individuals (e.g., trait level differences). The manner in which these variables impact pattern completion processes at retrieval, or make independent contributions to episodic remembering in older adults, is an important direction for future work. Nevertheless, the present results provide compelling initial evidence that (a) hippocampal and cortical indices of pattern completion play a central role in determining whether individual events will be remembered or forgotten, (b) that predicted relationships between hippocampal activity, reinstatement strength, and associative memory retrieval can be observed even late in the lifespan, and (c) and that these neural metrics explain unique variance in memory performance across individuals.

Hippocampal and cortical indices of pattern completion not only explained variance in our primary associative memory measures, but also in delayed recall performance on standardized neuropsychological tests — among the most widely used assays of episodic memory in the study of aging and disease. The relationship between these measures, collected during separate testing sessions, suggests that the neural indices derived from task-based fMRI are tapping into stable individual differences, and may represent a sensitive biomarker of hippocampal and cortical function. Critically, we also demonstrate that these neural and neuropsychological test measures explained unique variance in associative memory, together accounting for 50% of the variance in exemplar-specific recall across individuals. This not only indicates that the present neural indices provide information that cannot be garnered from paper and pencil tests alone, but also suggests that we can combine these neural metrics with existing measurement tools to build more accurate models to explain individual differences in memory performance in older adults. An important direction for future work is to assess whether combining task-related neural measures, such as those identified here, with other known biomarkers of brain health and disease risk (e.g., in vivo measures of amyloid and tau accumulation, hippocampal volume, white matter integrity; *Hedden et al., 2016*; *Jack et al., 2018*) can further increase sensitivity for explaining individual differences in memory performance, as well as predicting future disease risk and memory decline prior to the emergence of clinical impairment.

Taken together, the present results significantly advance our understanding of fundamental retrieval processes supporting episodic memory in cognitively normal older adults. By exploring how neural indices of pattern completion vary — both across trials and across individuals — these findings demonstrate that hippocampal activity and cortical reinstatement during memory retrieval provide a partial account for why and when older adults remember, and they predict which older adults will perform better than others across multiple widely adopted assays of episodic memory. They also suggest that some neural indices of pattern completion may be affected by age to a greater degree than others, though we note that both the presence and absence of age effects must be interpreted with caution due to the cross-sectional nature of the study design, and should be confirmed in the context of longitudinal studies. Moreover, we note that because the current sample is cognitively healthy, future work is needed to determine if similar patterns of results are observed across qualitatively different cohorts of older adults, particularly those in which subjective or mild cognitive decline is already apparent. Nevertheless, our findings underscore the striking heterogeneity in brain and behaviour, even among cognitively normal older adults, and lend support to the hypothesis that this high within-group variance likely contributes to the wealth of mixed findings in the literature, particularly for traditional group-level comparisons in the context of small-to-moderate sample sizes. Collectively, our findings illustrate how an individual differences approach can advance understanding of the neurocognitive mechanisms underlying variability in episodic memory in older adults.

## Materials and methods

### Participants

One hundred and five cognitively healthy older adults (aged 60–82 years; 65 female) participated as part of the Stanford Aging and Memory Study. Eligibility included: normal or corrected-to-normal vision and hearing; right-handed; native English speaking; no history of neurological or psychiatric disease; a Clinical Dementia Rating score of zero (CDR; *Morris, 1993*) and performance within the normal range on a standardized neuropsychological assessment (see *Neuropsychological Testing*). Data collection spanned multiple visits: Neuropsychological assessment was completed on the first visit and the fMRI session occurred on the second visit, with the exception of nine participants who completed the fMRI session on the same day as the neuropsychological testing session. Visits took place ~6.18 weeks apart on average (range = 1–96 days). Participants were compensated $50 for the clinical assessment and $80 for the fMRI session. All participants provided informed consent in accordance with a protocol approved by the Stanford Institutional Review Board. Data from five participants were excluded from all analyses due to excess head motion during scanning (see *fMRI preprocessing*), yielding a final sample of 100 older adults (60–82 years; 61 female; see *Table 1* for demographics).

### Neuropsychological testing

Participants completed a neuropsychological test battery consisting of standardized tests assessing a range of cognitive functions, including episodic memory, executive function, visuospatial processing, language, and attention. Scores were first reviewed by a team of neurologists and neuropsychologists to evaluate cognition and reach a consensus assessment that each participant was cognitively healthy, defined as performance on each task within 1.5 standard deviations of demographically adjusted means. Subsequently, a composite delayed recall score was computed for each participant by (a) z-scoring the delayed recall subtest scores from the Logical Memory (LM) subtest of the Wechsler Memory Scale, 3rd edition (WMS-III; *Wechsler, 1997*), Hopkins Verbal Learning Test-Revised (HVLT-R; *Brandt, 1991*), and the Brief Visuospatial Memory Test-Revised (BVMT-R; *Benedict, 1997*), and (b) then averaging. This composite score declined with age ($\beta = -0.21$, $p < 0.005$), was lower in males than females ($\beta = -0.35$, $p < 0.05$), but did not vary with years of education ($\beta = 0.07$, $p > 0.31$).

### Materials

Stimuli comprised words paired with colour photos of faces and scenes obtained from online sources. For each participant, 120 words (out of 150 words total) were randomly selected and paired with the pictures (60 word-face; 60 word-place) during a study phase, and these 120 words plus the remaining 30 words (foils) appeared as cues during the retrieval phase. Words were concrete nouns (e.g., 'banana', 'violin') between 4 and 8 letters in length. Faces corresponded to famous people (e.g., 'Meryl Streep', 'Ronald Reagan') and included male and female actors, musicians, politicians, and scientists. Places corresponded to well-known locations (e.g., 'Golden Gate Bridge', 'Niagara Falls') and included manmade structures and natural landscapes from a combination of domestic and international locations.

### Behavioural procedure

Prior to scanning, participants completed a practice session that comprised an abbreviated version of the task (12 word-picture pairs not included in the scan session). This ensured that participants understood the task instructions and were comfortable with the button responses. Participants had the option to repeat the practice round multiple times if needed to grasp the instructions.

Next, concurrent with fMRI, participants performed an associative memory task consisting of five rounds of alternating encoding and retrieval blocks (*Figure 1*). In each encoding block, participants viewed 24 word-picture pairs (12 word-face and 12 word-place) and were asked to intentionally form an association between each word and picture pair. To ensure attention to the pairs, participants were instructed to indicate via button press whether they were able to successfully form an association between items in the pair. Following each encoding block, participants performed a retrieval task that probed item recognition and associative recollection. In each block, 24 target words were

interspersed with 6 novel (foil) words; participants made a 4-way memory decision for each word. Specifically, if they recognized the word and recollected the associated image, they responded either 'Face' or 'Place' to indicate the category of the remembered image; if they recognized the word but could not recollect sufficient details to categorize the associated image, they responded 'Old'; if they did not recognize the word as studied, they responded 'New'. Responses were made via right-handed button presses, with four different finger assignments to the response options counterbalanced across participants. Using MATLAB Psychophysics Toolbox (*Brainard, 1997*), visual stimuli were projected onto a screen and viewed through a mirror; responses were collected through a magnet-compatible button box.

During both encoding and retrieval blocks, stimuli were presented for 4 s, followed by an 8 s inter-trial fixation. During retrieval blocks, the probe word changed from black to green text when there was 1 s remaining, indicating that the end of the trial was approaching and signaling participants to respond (if they had not done so already). After the MR scan session, a final overt cued-recall test was conducted outside the scanner to evaluate the degree to which participants were able to recollect the specific face or place associated with each target word. On this post-test, participants were presented with studied words, in random order, and asked to provide the name of the associate or, if not possible, a description of the associate in as much detail as they could remember. The post-test was self-paced, with responses typed out on a keyboard; participants were instructed to provide no response if no details of the associate could be remembered.

## Memory response classification

The fMRI retrieval trials were classified into six conditions: associative hits (studied words for which the participant indicated the correct associate category), associative misses (studied words for which the participant indicated the incorrect associate category), item hits (studied words correctly identified as 'old'), item misses (studied words incorrectly identified as 'new'), item false alarms (foils incorrectly called 'old'), associative false alarms (foils incorrectly indicated as associated with a 'face' or a 'place'), and correct rejections (CR; foils correctly identified as 'new'). Because the number of false alarms was low (M = 5.1, SD = 4.7), these trials were not submitted to fMRI analysis (see *Supplementary file 1* for a summary of trial counts and retrieval reaction time by memory outcome).

In-scanner associative memory performance was estimated using a discrimination index, associative *d'*. Hit rate was defined as the rate of correct category responses to studied words (associative hits) and the false alarm rate was defined as the rate of incorrect associative responses to novel words (associative false alarms). Thus, associative *d'* = Z('Correct Associate Category' | Old) – Z('Associate Category' | New). We additionally calculated an old/new discrimination index to assess basic understanding of and ability to perform the task. Here, hit rate was defined as the rate of correct old responses to studied words, irrespective of associative memory (associative hits, associative misses, item hits), and the false alarm rate was defined as the rate of incorrect old responses to novel words (item false alarms, associative false alarms). Thus, old/new *d'* = Z('Old' + 'Face' + 'Place' | Old) – Z('Old' + 'Face' + 'Place' | New).

The post-test data were analysed using a semi-automated method. Participants' typed responses were first processed with in house R code to identify exact matches to the name of the studied image. Responses that did not include exact matches were flagged, and subsequently assessed by a human rater, who determined the correspondence between the description provided by the participant and the correct associate. We computed the proportion of studied words for which the associate was correctly recalled (Exemplar Correct/All Old). One participant did not complete the post-test, leaving 99 participants in all analyses of the post-test data.

## MRI data acquisition

Data were acquired on a 3T GE Discovery MR750 MRI scanner (GE Healthcare) using a 32-channel radiofrequency receive-only head coil (Nova Medical). Functional data were acquired using a multi-band EPI sequence (acceleration factor = 3) consisting of 63 oblique axial slices parallel to the long axis of the hippocampus (TR = 2 s, TE = 30 ms, FoV = 215 mm x 215 mm, flip angle = 74, voxel size = 1.8 × 1.8 × 2 mm). To correct for B0 field distortions, we collected two B0 field maps before every functional run, one in each phase encoding direction. Two structural scans were acquired: a whole-brain high-resolution T1-weighted anatomical volume (TR = 7.26 ms, FoV = 230 mm × 230

mm, voxel size = 0.9 × 0.9 x 0.9 mm, slices = 186), and a T2-weighted high-resolution anatomical volume perpendicular to the long axis of the hippocampus (TR = 4.2 s, TE = 65 ms, FOV = 220 mm, voxel size = 0.43 × 0.43×2 mm; slices = 29). The latter was used for manual segmentation of hippocampal subfields and surrounding cortical regions (*Olsen et al., 2009*).

## fMRI preprocessing

Data were processed using a workflow of FSL (*Smith et al., 2004*) and Freesurfer (*Dale et al., 1999*) tools implemented in Nipype (*Gorgolewski et al., 2011*). Each timeseries was first realigned to its middle volume using normalized correlation optimization and cubic spline interpolation. To correct for differences in slice acquisition times, data were temporally resampled to the TR midpoint using sinc interpolation. Finally, the timeseries data were high-pass filtered with a Gaussian running-line filter using a cutoff of 128 s. The hemodynamic response for each trial was estimated by first removing the effects of motion, trial artifacts, and session from the timeseries using a general linear model. The residualized timeseries was then reduced to a single volume for each trial by averaging across TRs 3–5 (representing 4–10 s post-stimulus onset), corresponding to the peak of the hemodynamic response function. To preserve the high resolution of the acquired data, the data were left unsmoothed.

Images with motion or intensity artifacts were automatically identified as those TRs in which total displacement relative to the previous frame exceeded 0.5 mm or in which the average intensity across the whole brain deviated from the run mean by greater than five standard deviations. Runs in which the number of artifacts identified exceeded 25% of timepoints, as well as runs in which framewise displacement exceeded 2 mm, were excluded. These criteria led to exclusion of data from five participants who exhibited excess head motion across runs, as well as exclusion of one study and test run from an additional participant. Across all included runs from 100 participants, an average of 2.4 (SD = 3.7) encoding phase volumes (1.7% of volumes) and 2.6 (SD = 4.2) retrieval phase volumes (1.5% of volumes) were identified as containing an artifact. Trials containing fMRI artifacts were excluded from all analyses. To control for potential residual effects of head motion on our primary variables of interest, we adjusted each variable of interest by mean framewise displacement using linear regression (see *Supplementary file 1* for a summary of motion effects).

Using Freesurfer, we segmented the T1-weighted anatomical volume at the gray-white matter boundary and constructed tessellated meshes representing the cortical surface (*Dale et al., 1999*). Functional data from each run were registered to the anatomical volume with a six degrees-of-freedom rigid alignment optimizing a boundary-based cost function (*Greve and Fischl, 2009*). Finally, runs 2–4 were resampled into the space of run 1 using cubic spline interpolation to bring the data into a common alignment. All analyses were thus performed in participant native space, avoiding normalization to a group template.

## Regions of interest

Our analyses focus specifically on hippocampal pattern completion processes — via hippocampal univariate activity and multivariate cortical reinstatement metrics — in the aging brain. Thus, analyses were conducted in three a priori regions of interest (ROIs), selected based on existing theoretical and empirical work to optimize the measurement of this process. Analyses of task-evoked univariate activity were focused on the hippocampus, whereas multivoxel pattern analyses were conducted in ventral temporal cortex (VTC) and angular gyrus (ANG), two cortical areas that have been reliably linked to cortical reinstatement in healthy younger adults (*Kuhl et al., 2013*; *Gordon et al., 2014*; *Kuhl and Chun, 2014*; *Favila et al., 2018*; *Lee et al., 2019*). All ROIs were bilateral and defined in participants' native space (*Figure 2*).

The hippocampal mask was defined manually using each participant's high-resolution T2-weighted structural image using established procedures (*Olsen et al., 2009*), and comprised the whole hippocampus (see *Figure 4—figure supplements 5–6* for analysis of hippocampal subfields). The VTC mask was composed of three anatomical regions: parahippocampal cortex, fusiform gyrus, and inferior temporal cortex. The fusiform gyrus and inferior temporal cortex masks were generated from each participant's Freesurfer autosegmentation volume using bilateral inferior temporal cortex and fusiform gyrus labels. These were combined with a manually defined bilateral parahippocampal cortex ROI, defined using established procedures (*Olsen et al., 2009*), to form the VTC mask. The

ANG ROI was defined by the intersection of the Freesurfer inferior parietal lobe label and the Default Network of the Yeo 7 network atlas (*Yeo et al., 2011*), defined on the Freesurfer average (fsaverage) cortical surface mesh. This intersection was used to confine the ROI to the inferior parietal nodes of the Default Mode Network, which predominantly encompasses ANG (*Favila et al., 2018*). To generate ROIs in participants' native space from the fsaverage space label, we used the approach detailed in Waskom and colleagues (*Voss et al., 2008*), which uses the spherical registration parameters to reverse-normalize the labels, and then converts the vertex coordinates of labels on the native surface into the space of each participant's first run using the inverse of the functional to anatomical registration. Participant-specific ROIs were then defined as all voxels intersecting the midpoint between the gray-white and gray-pial boundaries.

## Multivoxel pattern classification

Our primary measure of category-level cortical reinstatement during memory retrieval was derived from multivoxel classification analysis. Classification was implemented using Scikit-learn (*Pedregosa et al., 2011*), nilearn (*Abraham et al., 2014*), nibabel (*Brett et al., 2016*), and in house Python scripts, and performed using L2-penalized logistic regression models as instantiated in the LIBLINEAR classification library (regularization parameter C = 1). These models were fit to preprocessed BOLD data from VTC and ANG that were reduced to a single volume for each trial by averaging across TRs 3–5. Prior to classification, the sample by voxel matrices for each region were scaled across samples within each run, such that each voxel had zero mean and unit variance. A feature selection step was also conducted, in which a subject-specific univariate contrast was used to identify the top 250 voxels that were most sensitive to each category (face, place) during encoding, yielding a set of 500 voxels over which classification analyses were performed. Prior to each of 10 iterations of classifier training, the data were subsampled to ensure an equal number of face and scene trials following exclusion of trials with artifacts.

To first validate that classification of stimulus category (face/place) during encoding was above chance for each ROI, we used a leave-one-run-out-n-fold cross-validation procedure on the encoding data. This yielded a value of probabilistic classifier output for each trial, representing the degree to which the encoding pattern for a trial resembled the pattern associated with a face or place trial. This output was converted to binary classification accuracy indicating whether or not a given test trial was correctly classified according to the category of the studied picture. Here we report the average classifier accuracy across folds for each participant in each ROI.

To measure category-level cortical reinstatement during memory retrieval, we trained a new classifier on all encoding phase data, and then tested on all retrieval phase data. For each retrieval trial, the value of probabilistic classifier output represented a continuous measure of the probability (range 0–1) that the classifier assigned to the relevant category for each trial (0 = certain place classification, 1 = certain face classification). For assessment of classifier performance across conditions (associative hits, associative misses, item only hits, and item misses) and ROI (VTC, ANG), we converted this continuous measure of classifier evidence to binary classification accuracy, indicating whether or not a given retrieval trial was correctly classified according to the category of the studied picture.

The significance of classifier performance for each condition and ROI was assessed using permutation testing. We generated a null distribution for each participant by shuffling the trial labels over 1000 iterations for each of the 10 subsampling iterations, calculating mean classifier accuracy for each iteration. We then calculated the mean number of times the permuted classifier accuracy met or exceeded observed classifier accuracy to derive a *p* value indicating the probability that the observed classifier accuracy could arise by chance.

For trial-wise analyses relating cortical reinstatement strength to memory behaviour (e.g., associative retrieval accuracy and reaction time) and other neural variables (e.g., hippocampal BOLD), a continuous measure of reinstatement strength was derived by calculating the logits (log odds) of the probabilistic classifier output on each trial. Reinstatement strength was signed in the direction of the correct associate for a given trial, such that, regardless of whether the trial was a face or place trial, the evidence was positive when the classifier guessed correctly, and negative when the classifier guessed incorrectly. The magnitude of reinstatement strength was thus neutral with respect to which associate category (face or place) was retrieved. For individual-differences analyses relating cortical reinstatement strength to age and memory behaviour (e.g., associative *d'*, exemplar-specific recall),

we computed the mean category-level reinstatement strength (i.e., logits) across associative hit trials for each participant.

## Pattern similarity analysis

To complement the classification analyses, we used pattern similarity analyses to measure event-level cortical reinstatement. This approach involved computing the similarity (Pearson correlation) between trial-wise activity patterns extracted from ROIs during encoding and retrieval (i.e., encoding-retrieval similarity; ERS). This analysis approach affords the opportunity to not only examine reinstatement at the categorical level (i.e., within-category ERS – between-category ERS) but also at the trial-unique item level (i.e., within-event ERS – within-category ERS). For this analysis, we again used the voxelwise activity patterns for each ROI (this time with no feature selection step), computing the correlation between encoding and retrieval patterns separately for successful (i.e., associative hits) and unsuccessful (i.e., associative misses, item only hits, item misses) retrieval trials, such that the events being compared (within-event, within-category, between-category) were matched on associative retrieval success. Within-category ERS was computed after values on the diagonal of the correlation matrices (i.e. within-event correlations) were removed, ensuring that event-level ERS does not contribute to the within-category ERS estimate. All correlations were Fisher transformed before computing the mean correlation between different events of interest.

## Statistical analysis

All statistical analyses were implemented in the R environment (version 3.4.4). Trial-wise analyses were conducted using mixed effects models (linear and logistic) using the lmer4 statistical package (*Bates et al., 2015*). Each model contained fixed effects of interest, a random intercept modeling the mean subject-specific outcome value, and a random slope term modeling the subject-specific effect of the independent variable of interest (e.g., hippocampal activity, reinstatement strength). Models also contained nuisance regressors (see *Supplementary file 1* for a full list of regressors in each model), including stimulus category, age, ROI encoding classifier strength (when reinstatement strength —logits— was the independent or dependent variable), ROI univariate activity in category-selective voxels (when reinstatement strength – logits – was the independent variable, controlling for activity in voxels identified during feature selection, over which classification was performed), overall ROI univariate activity (when ERS was the independent variable, controlling for activity in the whole ROI, as no feature selection step was conducted for pattern similarity analyses), and category-level ERS (when event-level ERS was the independent or dependent variable, to mitigate the possibility that effects of event-level ERS can be attributed to category-level reinstatement). Models were conducted over all test trials in which a studied item was presented, except where indicated that only associative hit trials were included (see *Supplementary file 1* for a summary of results when item miss trials are excluded from analyses). Random slopes were uncorrelated from random intercepts to facilitate model convergence. The significance of effects within mixed-model regressions was obtained using log-likelihood ratio tests, resulting in $\chi^2$ values and corresponding *p*-values. A Wald *z*-statistic was additionally computed for model parameters to determine simultaneous significance of coefficients within a given model. All continuous variables were z-scored within participant across all trials prior to analysis. For trial-wise mediation analyses, the coefficient of the indirect path was computed as the product of the direct effects, $a \times b$. The significance of the indirect effect was calculated with bootstrap resampling with 5000 iterations of data sampled with replacement, and was considered significant if zero does not fall within the 95% confidence interval of the bootstrapped estimate of the indirect effect; 95% confidence intervals are reported.

Individual-differences analyses were conducted using multiple linear regression. In all regression models, each neural variable was computed by taking the mean value over associative hit trials. For hippocampal BOLD, mean activity during associative hits was corrected by subtracting mean activity during correct rejections. Before entry into regression models, each neural variable was further adjusted by head motion (mean framewise displacement) and, in the cases of reinstatement strength, ROI-specific encoding classifier strength (mean logits) to account for individual differences in category differentiation during encoding. Age-independent models adjusted memory scores by age. Main text figures depict raw values for interpretability (see *Figure 5—figure supplement 1* for partial plots). Hierarchical regression was used to assess the relative contributions of each

independent variable to memory performance. F ratio statistics were used to determine change in explained variance ($R^2$) at each step compared to the previous step. The explanatory power of each regression model was evaluated descriptively using the explained variance (adjusted $R^2$). All continuous variables were z-scored across participants prior to analysis, producing standardized coefficients. All analyses used a two-tailed level of 0.05 for defining statistical significance.

## Acknowledgements

The Stanford Aging and Memory Study (SAMS) is supported by the National Institute on Aging (R01AG048076, R21AG058111, and R21AG058859), Stanford's Center for Precision Health and Integrated Diagnostics (PHIND), and Stanford's Wu Tsai Neurosciences Institute. We are grateful to: Adam Kerr, Hua Wu, Michael Perry, and Laima Baltusis at the Stanford's Center for Cognitive and Neurobiological Imaging (CNI) for their assistance in fMRI data acquisition; Clementine Chou, Madison Kist, and Austin Salcedo for their assistance with data collection; and the SAMS volunteers for their participation in the study.

## Additional information

### Funding

| Funder | Grant reference number | Author |
| --- | --- | --- |
| National Institute on Aging | R01AG048076 | Anthony D Wagner |
| National Institute on Aging | R21AG058111 | Anthony D Wagner |
| National Institute on Aging | R21AG058859 | Elizabeth C Mormino |
| Stanford Center for Precision Health and Integrated Diagnostics | PHIND | Brian K Rutt<br>Elizabeth C Mormino<br>Anthony D Wagner |
| Stanford Wu Tsai Neuroscience Institute | Seed Grant | Elizabeth C Mormino<br>Anthony D Wagner |

The funders had no role in study design, data collection and interpretation, or the decision to submit the work for publication.

### Author contributions

Alexandra N Trelle, Conceptualization, Data curation, Software, Formal analysis, Validation, Investigation, Visualization, Methodology, Writing - original draft, Writing - review and editing; Valerie A Carr, Conceptualization, Formal analysis, Funding acquisition, Investigation, Methodology, Writing - review and editing; Scott A Guerin, Conceptualization, Data curation, Formal analysis, Supervision, Investigation, Methodology, Project administration, Writing - review and editing; Monica K Thieu, Data curation, Formal analysis, Investigation, Project administration, Writing - review and editing; Manasi Jayakumar, Wanjia Guo, Ayesha Nadiadwala, Nicole K Corso, Madison P Hunt, Celia P Litovsky, Natalie J Tanner, Jeffrey D Bernstein, Data curation, Investigation, Project administration; Gayle K Deutsch, Sharon J Sha, Carolyn A Fredericks, Supervision, Investigation; Marc B Harrison, Data curation, Validation, Investigation; Anna M Khazenzon, Validation, Investigation; Jiefeng Jiang, Software, Validation; Brian K Rutt, Resources, Funding acquisition, Methodology; Elizabeth C Mormino, Supervision, Funding acquisition, Project administration, Writing - review and editing; Geoffrey A Kerchner, Conceptualization, Funding acquisition, Methodology; Anthony D Wagner, Conceptualization, Resources, Supervision, Funding acquisition, Methodology, Project administration, Writing - review and editing

### Author ORCIDs

Alexandra N Trelle [ID] https://orcid.org/0000-0003-2837-8753
Anthony D Wagner [ID] http://orcid.org/0000-0003-0624-4543

### Ethics

Human subjects: All participants provided informed consent in accordance with a protocol approved by the Stanford Institutional Review Board (IRB #30218).

### Decision letter and Author response

Decision letter https://doi.org/10.7554/eLife.55335.sa1
Author response https://doi.org/10.7554/eLife.55335.sa2

## Additional files

### Supplementary files

• Supplementary file 1. Supplementary tables. Supplementary file 1a. Neuropsychological test battery performance. Supplementary file 1b. Reaction time (ms) and trial counts as a function of trial type. Supplementary file 1c. Summary of model parameters for mixed effects models. Supplementary file 1d. Summary of linear and logistic mixed effects model results when item miss trials are excluded. Supplementary file 1e. Summary of linear and logistic mixed effects models examining effects of stimulus category (face, place) on relationships between neural variables and behavioural variables. Supplementary file 1f. Summary of linear mixed effects models examining effects of stimulus category (face, place) on relationships between hippocampal activity and cortical reinstatement. Supplementary file 1g. Analysis of head motion and its effects on key dependent variables of interest. Supplementary file 1h. Summary of hierarchical regression analysis predicting associative d'. Supplementary file 1i. Summary of regression analyses examining the relationship between hippocampal subfield activity during associative retrieval (associative hit - CR) and associative memory.

• Transparent reporting form

### Data availability

Source data files have been provided for Table 1 and Figures 3-5. Data and code for reproducing all analyses, results, and figures in the paper are available at https://github.com/alitrelle/sams_hpc_fmri (copy archived at https://github.com/elifesciences-publications/sams_hpc_fmri).

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
