## [Decision Letter]

**Acceptance summary:**

This paper combines an elegant cognitive task with sophisticated brain imaging approaches (functional MRI pattern analysis) to characterize the neural mechanisms that support memory retrieval in healthy older adults. While it is well known that memory declines with aging, our understanding of how neural variability – both across individuals and within an individual, on a moment-to-moment basis – might explain memory success or failure remains limited. Armed with a large sample size well-suited to address just these sorts of questions, the present study provides clear empirical evidence for unique hippocampal and cortical contributions to remembering in healthy older adults.

**Decision letter after peer review:**

Thank you for submitting your article "Hippocampal and cortical mechanisms at retrieval explain variability in episodic remembering in older adults" for consideration by *eLife*. Your article has been reviewed by three peer reviewers, one of whom is a member of our Board of Reviewing Editors, and the evaluation has been overseen by Laura Colgin as the Senior Editor. The following individuals involved in review of your submission have agreed to reveal their identity: Lars Nyberg (Reviewer #2); Josh Koen (Reviewer #3).

The reviewers have discussed the reviews with one another and the Reviewing Editor has drafted this decision to help you prepare a revised submission.

Summary:

In this interesting and well-written study, the authors used functional magnetic resonance imaging to ask whether hippocampal pattern completion can account for some of the marked heterogeneity in episodic memory decline in healthy aging. Pattern completion was indexed with multivariate measures of cortical reinstatement (category- and event-level) and univariate hippocampal activity. Importantly, trial-by-trial measures of pattern completion in the ANG and VTC we related to successful associative memory retrieval and also account for some of the age-related declines in memory performance. Reviewers appreciated the sizeable sample of older adult participants, and highlighted the consideration of within-participant, trial-by-trial neural effects was a notable methodological advance. This paper will be an important contribution to the field for these reasons.

Essential revisions:

While reviewers are overall enthusiastic about this paper, there are a number of concerns identified that must be adequately addressed in a revision.

1) The reporting of the linear mixed effects models needs to be clearer in the paper. It was quite challenging to keep track of what predictors were included in each model when they were reported in the text. For example, please clarify what the "relevant nuisance variables" are and why they were deemed "relevant" for each model. One way to help this is to predictors in each model and their associated model coefficients in supplementary materials (including the random effects components). This has the added benefits of providing more information to readers to better evaluate the results. The reviewers acknowledge that this may be redundant with information on the GitHub repository (which was appreciated!), but note that not all readers will access that resource.

2) Reviewers were confused by reporting what are essentially the same models twice, with one set having age moderate the effect. This is particularly problematic as some of the effects in the first part of the results are actually moderated by age (for example, there was an Age Reinstatement interaction predicting AH success). Significant interactions with age make the earlier results from models that did not have an age interaction term more difficult to interpret. Reviewers request that the authors consider including only the larger models with the age-by-predictor of interest terms in the main paper, reporting whether the age-by-fixed effect interaction terms were significant or not. Additional models might be better put into the supplementary material if the authors deem them necessary to report. Reviewers believe this will improve the overall clarify of the statistical approach.

3) There are many known issues with interpreting mediation from cross-sectional samples (see Raz and Lindenburger, 2011, Psychological Bulletin; Maxwell and Cole, 2007, Psychological Methods), and thus the conclusions one can draw from them here are limited. Here, we note that the mediation analyses were not critical to the paper's main conclusions. As such, the reviewers recommend removing the mediation analyses to improve the flow of the paper and reduce the already large number of analyses. However, we acknowledge that this issue might still be debated in the field and that mediation analyses are regularly reported in cross-sectional studies. For this reason, reviewers believe it is the authors' right to decide the best course of action for their paper. If the authors elect to keep the mediation analysis, please add more details about it to the Materials and methods and discuss its limitations in a cross-sectional study.

4) There is growing appreciate in the literature about how fidelity of representation at encoding might explain some of the observed age differences in retrieval (at the behavioral or neural levels; for recent reviews, see Koen and Rugg, 2019, TICs; Koen et al., 2020, Current Opinion in Behavioral Sciences). As such, it would be appropriate to include more about the simple relationships between encoding strength and the various retrieval measures. The authors state that encoding strength is correlated with reinstatement strength and subsequently include it as a covariate of no interest in their statistical models. I understand that this demonstrates that action at retrieval is an additional and important contributor. However, it seems as though more information on encoding strength on its own is warranted. Please report whether there are age effects on the encoding classification data. As I understand it, encoding strength does not "explain away" the age effects observed here – is that correct? At the conceptual level, I believe the relationship among hippocampal activity, reinstatement, and memory success should be intact even when encoding strength is weak, but perhaps this has a larger effect on the reinstatement piece (i.e., less on hippocampal activity)? While we understand the encoding classification data is not central to the aims of this paper, it deserves some mention to show that this task can replicate prior findings showing reductions in multivariate measures of neural selectivity (see Carp et al., 2011; Park et al., 2010; 2012; Koen et al., 2019). These papers also warrant citing in the relevant spots when discussing neural dedifferentiation.

5) As a methodological point (somewhat related to the possibility of dedifferentiation), how might encoding strength influence reinstatement at the level of activation patterns and correlations? In other words, if there is additional noise in the patterns at encoding, one might expect that to show up in subsequent analyses that also depend on these patterns (this would be true both for a classifier trained on the encoding data or the ERS analysis). This seems worth consideration in the paper, even if the authors do not believe it to be a relevant issue.

6) It is stated (Introduction) that aging may affect HPC, cortical reinstatement, and/or the brain-behavior link. While reviewers agree in principle, in view of the canonical model outlined in the 1st paragraph of the Introduction (cue => HC activation => cortical replay), it is unclear whether intact cortical reinstatement in principle could occur in the presence of dysfunctional HPC. The authors may want to elaborate on this. It is unclear whether the regression findings reported in the subsection “Unique Hippocampal and Cortical Contributions to Associative Retrieval” offer conclusive support that VTC and ANG can support successful associative retrieval *independently* of hippocampal activity (i.e. that they would be predictive in the absence of HC activity). This topic is also critical for the conclusion that age particularly affected the translation of cortical evidence to memory behavior (subsection “Effects of Age on Hippocampal and Cortical Indices of Pattern Completion”). The finding that HPC may be preserved but cortical reinstatement impaired in ageing is quite surprising in view of findings of preserved retrieval of detail semantic memories in older age, along with impairment on hippocampus-based episodic memory tasks. Please address this, perhaps in the Discussion.

7) Given the ideas that hippocampal activity should drive cortical reinstatement, one might expect the authors would test whether reinstatement (perhaps separately in VTC and ANG) mediates the relationship between hippocampal activity and associative memory success. Instead, the authors asked whether both explained some variance in performance when included in the same model. This seems to make a slightly different statistical point. It would be helpful for the authors to include some rationale for their decision of statistical test in light of the mechanism.

8) The analyses of cortical reinstatement focused on VTC and ANG, with the motivation that these areas support content-rich representations during retrieval and that their representations may be related to memory-guided behaviour. The VTC prediction was easy to follow, given face/place selectivity in this area. However, reviewers wanted more information on the role of ANG as an a priori region of interest. What type of processing is this region thought to perform? For example, would ANG involvement be expected if auditory information is studied rather than visual objects? That is, is this a region related to stimulus specificity or more generally related to successful retrieval ("memory guided behaviour")? If the latter, is it really to be expected to see "reactivation"/reinstatement of encoding activity in this region?

9) More justification is needed for grouping AM, IH, and IM into a single unsuccessful bin logistic mixed effects regression model. Doing so makes it difficult to know if the models are predicting associative memory success or simply item memory success. The more appropriate comparison might be to drop IM trials from this bin such that the AH vs. AM + IH controls for successful item memory. It would be important for the authors to somehow demonstrate that the results do not hinge on item miss trials being included in the "unsuccessful" bin. This analysis could be reported in the supplementary material. Also, it would be helpful to report the proportions of trials in each bin in supplementary material for readers to appreciate the relative mixture of trial types.

10) Some of the effects appear to depend on image category (according to Supplementary file 1E). This seems rather important to touch on in the main text. There is some inconsistency in the literature concerning if age-related neural dedifferentiation is ubiquitous or not across different visual stimuli (Voss et al., 2008; Koen et al., 2019; Payer et al., 2006; reviewed in Koen and Rugg, 2019; Koen et al., 2020). Thus, these category moderations deserve some mention in the main text. Please also address whether there are behavioral (memory) differences between faces and places, since substantial differences in performance across these trial types could influence the classification, as memory success/failure is confounded with face/place.

11) Please clarify how the fMRI measures were summarized for the individual difference analyses. Were they simply averaged across all trials, AH trials, or some other mixture? Also, why were age interactions excluded from the model? While the models suggest that the effects are independent of age, they do not imply that they are age invariant (Rugg, 2015). I appreciate that there might not be enough variability in this sample (which is only older adults) but demonstrating age invariance here seems important given the conclusions that are drawn.

12) The degree of variability in any given sample/study will in part be driven by the recruitment procedure. It is our impression that this sample was very healthy and cognitively intact (see Table 1) – and there was no relation between level of education and memory performance! This is at odds with findings in many published longitudinal studies. Comments on the sample in relation to the topic of heterogeneity would be informative.

13) Effects now attributed to influences of ageing in this cross-sectional study could be driven by other factors, and strong conclusions on the effects go ageing will have to await longitudinal follow-up studies. Please address this limitation in the paper.

[Editors' note: further revisions were suggested prior to acceptance, as described below.]

Thank you for resubmitting your work entitled "Hippocampal and cortical mechanisms at retrieval explain variability in episodic remembering in older adults" for further consideration by *eLife*. Your revised article has been evaluated by Laura Colgin as the Senior Editor and a Reviewing Editor.

The manuscript has been improved but there are some remaining issues that need to be addressed before acceptance, as outlined below:

1) Please clarify throughout the paper, figures (i.e., axis labels), and legends when the term "reinstatement strength" refers to category-level versus event-level reinstatement.

2) Please clarify what is meant by "univariate activity" in Supplementary file 1C. Are all univariate measures derived from hippocampus, or was activity from other ROIs (VTC, ANG) also included in some models as a control? Furthermore, what is the reason for sometimes using category-selective voxels and other times using whole-ROI activity? These details could be noted in the paper or table caption.

---

## [Author Response]

Essential revisions:While reviewers are overall enthusiastic about this paper, there are a number of concerns identified that must be adequately addressed in a revision.1) The reporting of the linear mixed effects models needs to be clearer in the paper. It was quite challenging to keep track of what predictors were included in each model when they were reported in the text. For example, please clarify what the "relevant nuisance variables" are and why they were deemed "relevant" for each model. One way to help this is to predictors in each model and their associated model coefficients in supplementary materials (including the random effects components). This has the added benefits of providing more information to readers to better evaluate the results. The reviewers acknowledge that this may be redundant with information on the GitHub repository (which was appreciated!), but note that not all readers will access that resource.

We appreciate the importance of enhanced clarity and transparency in reporting of the linear mixed effects models. Following the reviewer’s guidance, we added a table that lists all predictors, including both fixed and random effects, for all linear and logistic mixed effect models (Supplementary file 1C). We also revised the language within the manuscript (Materials and methods subsection “Statistical Analysis”) to increase clarity regarding nuisance variables and point readers to Supplementary file 1C.

2) Reviewers were confused by reporting what are essentially the same models twice, with one set having age moderate the effect. This is particularly problematic as some of the effects in the first part of the results are actually moderated by age (for example, there was an Age Reinstatement interaction predicting AH success). Significant interactions with age make the earlier results from models that did not have an age interaction term more difficult to interpret. Reviewers request that the authors consider including only the larger models with the age-by-predictor of interest terms in the main paper, reporting whether the age-by-fixed effect interaction terms were significant or not. Additional models might be better put into the supplementary material if the authors deem them necessary to report. Reviewers believe this will improve the overall clarify of the statistical approach.

We thank the reviewers for pointing out this potentially confusing and redundant description of models. Following their guidance, we revised the Results section to describe only the larger models that include the age-by-predictor of interest effects.

3) There are many known issues with interpreting mediation from cross-sectional samples (see Raz and Lindenburger, 2011, Psychological Bulletin; Maxwell and Cole, 2007, Psychological Methods), and thus the conclusions one can draw from them here are limited. Here, we note that the mediation analyses were not critical to the paper's main conclusions. As such, the reviewers recommend removing the mediation analyses to improve the flow of the paper and reduce the already large number of analyses. However, we acknowledge that this issue might still be debated in the field and that mediation analyses are regularly reported in cross-sectional studies. For this reason, reviewers believe it is the authors' right to decide the best course of action for their paper. If the authors elect to keep the mediation analysis, please add more details about it to the Materials and methods and discuss its limitations in a cross-sectional study.

We thank the reviewers for drawing our attention to these conceptual discussions regarding mediation analyses in cross-sectional designs. We appreciate and agree with the challenges inherent to interpreting mediation in this context. As the reviewers note, the mediation analysis is not central to the study’s aims or results; following their guidance, we removed this analysis from the paper.

4) There is growing appreciate in the literature about how fidelity of representation at encoding might explain some of the observed age differences in retrieval (at the behavioral or neural levels; for recent reviews, see Koen and Rugg, 2019, TICs; Koen et al., 2020, Current Opinion in Behavioral Sciences). As such, it would be appropriate to include more about the simple relationships between encoding strength and the various retrieval measures. The authors state that encoding strength is correlated with reinstatement strength and subsequently include it as a covariate of no interest in their statistical models. I understand that this demonstrates that action at retrieval is an additional and important contributor. However, it seems as though more information on encoding strength on its own is warranted. Please report whether there are age effects on the encoding classification data. As I understand it, encoding strength does not "explain away" the age effects observed here – is that correct? At the conceptual level, I believe the relationship among hippocampal activity, reinstatement, and memory success should be intact even when encoding strength is weak, but perhaps this has a larger effect on the reinstatement piece (i.e., less on hippocampal activity)? While we understand the encoding classification data is not central to the aims of this paper, it deserves some mention to show that this task can replicate prior findings showing reductions in multivariate measures of neural selectivity (see Carp et al., 2011; Park et al., 2010; 2012; Koen et al., 2019). These papers also warrant citing in the relevant spots when discussing neural dedifferentiation.

Thank you for raising these points. As summarized by the reviewers, we reiterate here that a) the focus of this paper is on processes at retrieval, b) our analytic approach of controlling for encoding effects demonstrates that action at retrieval is an additional and important contributor, and c) controlling for encoding strength does not explain away the age effects we observe on cortical reinstatement strength. Importantly, we agree that effects at retrieval (especially reinstatement strength) will in part reflect effects at encoding (as noted in Results subsection “fMRI Encoding Classifier Accuracy”), and this motivated our analytic approach to control for encoding strength in all analyses.

The above said, given the reviewers’ guidance to further highlight relevant findings at encoding that are helpful for interpreting the effects at retrieval that are the focus of this paper, we implemented the following additional edits:

a) We moved the statistics describing the relationship (both trial-wise, within-subject and individual differences, across-subject) between encoding strength and reinstatement strength to the main text (Results subsection “fMRI Encoding Classifier Accuracy”). These relationships are also depicted in Figure 5—figure supplement 3.

b) We added statistics describing the effect of age on encoding strength to the main text (Results subsection “fMRI Encoding Classifier Accuracy”).

c) We added a table listing all predictors for each model (Supplementary file 1C), making it easier to keep track of which models control for encoding strength. We also described this analytic approach in greater detail in the main manuscript (Results subsection “fMRI Encoding Classifier Accuracy”; Materials and methods- subsection “Statistical Analysis”). To reiterate here, encoding strength is included in all models in which a) reinstatement strength is a dependent variable, and b) reinstatement strength is the primary predictor of interest.

d) We discuss variance in differentiation/selectivity at encoding as it relates to processes at retrieval and memory performance (Discussion, sixth and seventh paragraphs), and added additional citations of relevant work per the reviewers’ guidance.

We note that the effect of age on encoding strength was not significant in VTC (*β* = -0.13, *p* = .133) or ANG (*β* = -0.06, *p* = .544). However, we do not believe this to be inconsistent with prior literature regarding neural dedifferentiation, nor does this result offer conclusive evidence regarding age effects at encoding. First, evidence for neural dedifferentiation with age comes from work contrasting effects between older and younger adults (e.g., Carp et al., 2011, Trelle et al., 2019, Koen et al., 2019) and when examining individual differences in differentiation across the lifespan (e.g., Park et al., 2012). Given the present manuscript’s focus on within- and between-subject variance using a cross-sectional design selectively focused on older adults, our study does not permit assessment of encoding differences between older and young adults. Second, our reported measure of encoding strength –– classifier evidence –– may be less sensitive to dedifferentiation than other methods, such as pattern similarity analysis, which was used in prior work examining this question (e.g., Carp et al., 2011, Trelle et al., 2019, Koen et al., 2019). We plan to report on the encoding data in this cohort in an independent manuscript, which will more thoroughly address these and related questions (addressing issues that are beyond the present manuscript’s focus on retrieval effects).

5) As a methodological point (somewhat related to the possibility of dedifferentiation), how might encoding strength influence reinstatement at the level of activation patterns and correlations? In other words, if there is additional noise in the patterns at encoding, one might expect that to show up in subsequent analyses that also depend on these patterns (this would be true both for a classifier trained on the encoding data or the ERS analysis). This seems worth consideration in the paper, even if the authors do not believe it to be a relevant issue.

As described in response to Point 4, our analytic approach is specifically designed to control for the possibility that variability with respect to noise in BOLD activity patterns at encoding (both across individuals and across trials) impacts subsequent patterns at retrieval, which we agree is a relevant issue. To make this point and the steps taken to mitigate it more clear in the manuscript, we made changes to the manuscript that are described in Response 4 (and can be found in the Results subsection “fMRI Encoding Classifier Accuracy”, Discussion, sixth and seventh paragraphs and in the Materials and methods subsection “Statistical Analysis”).

6) It is stated (Introduction) that aging may affect HPC, cortical reinstatement, and/or the brain-behavior link. While reviewers agree in principle, in view of the canonical model outlined in the 1st paragraph of the Introduction (cue => HC activation => cortical replay), it is unclear whether intact cortical reinstatement in principle could occur in the presence of dysfunctional HPC. The authors may want to elaborate on this. It is unclear whether the regression findings reported in the subsection “Unique Hippocampal and Cortical Contributions to Associative Retrieval” offer conclusive support that VTC and ANG can support successful associative retrieval independently of hippocampal activity (i.e. that they would be predictive in the absence of HC activity). This topic is also critical for the conclusion that age particularly affected the translation of cortical evidence to memory behavior (subsection “Effects of Age on Hippocampal and Cortical Indices of Pattern Completion”). The finding that HPC may be preserved but cortical reinstatement impaired in ageing is quite surprising in view of findings of preserved retrieval of detail semantic memories in older age, along with impairment on hippocampus-based episodic memory tasks. Please address this, perhaps in the Discussion.

This is an important theoretical issue, and we appreciate the reviewers highlighting the need for further clarity surrounding this core hypothesis and interpretation of key results. First, we clarify that we do not believe that cortical reinstatement in ANG and VTC could occur independently of hippocampal activity, or in the absence of a functioning hippocampus. Indeed, much evidence from studies in nonhuman animals documents the critical role of the hippocampus in the initiation of cortical reinstatement (e.g., Nakazawa et al., 2002; Tanaka et al., 2014). Instead, we intended to communicate that hippocampal activity and reinstatement strength explain unique variance in the probability of associative retrieval (i.e., they are not redundant predictors of memory behaviour). We revised the language in the Results (subsection “Cortical Reinstatement Partially Mediates the Effect of Hippocampal Activity on Retrieval”) to better reflect this. That is, although reinstatement is initiated by hippocampal activity (consistent with the observed relationship between these measures in the present sample), multivoxel patterns reinstated in cortex appear to carry additional information relevant for behaviour that is not contained in the univariate magnitude of hippocampal activity alone. One possibility is that this includes information about the specificity or precision with which event features are recovered (e.g., vividly recalling an image of the golden gate bridge vs. recalling that the image was a landmark but unsure of which one). Were this the case, it would suggest that some variability in the precision of reinstatement stems from factors beyond hippocampal pattern completion. A second, not mutually exclusive possibility is that this information is influenced by top-down post-retrieval monitoring, selection, and/or decision processes at retrieval, which further alter the extent to which retrieval patterns (and perhaps their precision) resemble encoding patterns and with relevance to memory behaviour. Critically, even if hippocampal activity is fixed (and non-zero), each of these factors could impact the magnitude of cortical reinstatement strength and influence memory decisions. We elaborate on these ideas in the Discussion (seventh paragraph) to clarify these important theoretical points. Taken together, we believe the present findings are not only compatible with computational models of pattern completion, but also highlight that there are additional influences –– beyond hippocampal pattern completion –– on cortical reinstatement strength and the mapping of cortical patterns to memory behaviour.

7) Given the ideas that hippocampal activity should drive cortical reinstatement, one might expect the authors would test whether reinstatement (perhaps separately in VTC and ANG) mediates the relationship between hippocampal activity and associative memory success. Instead, the authors asked whether both explained some variance in performance when included in the same model. This seems to make a slightly different statistical point. It would be helpful for the authors to include some rationale for their decision of statistical test in light of the mechanism.

We appreciate the reviewers highlighting this important distinction, and agree that a formal test of whether cortical reinstatement strength mediates the relationship between hippocampal activity and associative memory success is more appropriate in this context. We ran this analysis, separately for VTC and ANG, and revised the Results (subsection “Cortical Reinstatement Partially Mediates the Effect of Hippocampal Activity on Retrieval”) accordingly. Briefly, the results indicate that the relationship between hippocampal activity and the probability of associative retrieval is partially mediated by reinstatement strength, and this is observed for both VTC and ANG. This finding further supports the interpretation that hippocampal activity and reinstatement strength each measure a component of hippocampal pattern completion processes, and also indicates that the information carried by each measure explains both common and unique variance in retrieval success. We thank the reviewers for this guidance, as this additional reporting strengthens the reported findings and clarifies their theoretical implications (as we now emphasize in the Discussion, second paragraph).

8) The analyses of cortical reinstatement focused on VTC and ANG, with the motivation that these areas support content-rich representations during retrieval and that their representations may be related to memory-guided behaviour. The VTC prediction was easy to follow, given face/place selectivity in this area. However, reviewers wanted more information on the role of ANG as an a priori region of interest. What type of processing is this region thought to perform? For example, would ANG involvement be expected if auditory information is studied rather than visual objects? That is, is this a region related to stimulus specificity or more generally related to successful retrieval ("memory guided behaviour")? If the latter, is it really to be expected to see "reactivation"/reinstatement of encoding activity in this region?

We appreciate this question, as it surfaces that we did not sufficiently motivate predictions about cortical reinstatement in ANG. The inclusion of ANG as an a priori ROI is based on a rich and growing literature documenting evidence for cortical reinstatement of both category- and stimulus/event-specific features in this region during memory retrieval (Kuhl, Johnson and Chun, 2013; Kuhl and Chun, 2014; Lee and Kuhl, 2016; Favila et al., 2016; Thakral et al., 2017; Xiao et al., 2017; Lee et al., 2018). While the majority of these studies used visual stimuli, as is common in episodic memory research, there also is evidence for reinstatement of audio-visual stimuli in ANG (Bonnici et al., 2016). This body of work provides evidence that ANG not only signals retrieval success (independent of content), but supports representations of event features. The specific role of ANG during memory retrieval remains a topic of much interest in the literature (for reviews, see: Wagner et al., 2005; Cabeza et al., 2008; Hutchinson et al., 2009; Shimamura, 2011; Rugg and King, 2017; Ramanan et al., 2018). Theoretical interpretations of these data include a specific role in representing goal-relevant features during memory retrieval, and cross-modal integration or binding of episodic features. In the revision, we clarified how the extant literature motivates an analytic focus on ANG (Introduction, last paragraph).

9) More justification is needed for grouping AM, IH, and IM into a single unsuccessful bin logistic mixed effects regression model. Doing so makes it difficult to know if the models are predicting associative memory success or simply item memory success. The more appropriate comparison might be to drop IM trials from this bin such that the AH vs. AM + IH controls for successful item memory. It would be important for the authors to somehow demonstrate that the results do not hinge on item miss trials being included in the "unsuccessful" bin. This analysis could be reported in the supplementary material. Also, it would be helpful to report the proportions of trials in each bin in supplementary material for readers to appreciate the relative mixture of trial types.

We appreciate that the current modelling approach could potentially be influenced by the inclusion of item misses, particularly if these trials comprised a large proportion of non-AH trials. To ensure that the item miss trials are not driving the observed effects, we reran all mixed effects models excluding IM trials and replicated the results. Following the reviewers’ guidance, we added Supplementary file 1D, that summarizes the results of these models. We also now report summary statistics describing the proportion of trials in each condition in Supplementary file 1B to increase transparency regarding the mixture of trial types.

10) Some of the effects appear to depend on image category (according to Supplementary file 1E). This seems rather important to touch on in the main text. There is some inconsistency in the literature concerning if age-related neural dedifferentiation is ubiquitous or not across different visual stimuli (Voss et al., 2008; Koen et al., 2019; Payer et al., 2006; reviewed in Koen and Rugg, 2019; Koen et al., 2020). Thus, these category moderations deserve some mention in the main text. Please also address whether there are behavioral (memory) differences between faces and places, since substantial differences in performance across these trial types could influence the classification, as memory success/failure is confounded with face/place.

We appreciate the potential moderating effect of image category on the present neural measures and behaviour, and believe it important to report any such effects for completeness and as a resource for future related work. Given the length and richness of the Results section, we believe it is best to report these findings in Supplementary file 1E in order to keep the main findings digestible and clear, as the effects of image category do not bear on the primary aims of the manuscript. Critically, category was included as a factor in all mixed effects models, and the moderating effects of image category, when observed, do not impact interpretability of any of the key findings. For example: (1) The relationship between hippocampal activity and retrieval success did not vary by image category, and this relationship was significant for both categories; (2) While the relationship between hippocampal activity and category reinstatement strength interacted with category, critically, this relationship was significant for both image categories; (3) Similarly, while the relationships between reinstatement strength and (a) retrieval success and (b) decision RT interacted with category, these effects were significant for both image categories. Thus, when interactions with image category are observed, the interactions reflect the relative strength of a given effect in each image category, rather than the presence of an effect within one category but not the other. There is one exception, the relationship between hippocampal activity and event-level reinstatement in VTC is significant on place trials but not face trials.

We implemented the following changes to increase awareness and accessibility of the effects of image category:

a) We now describe the effect of image category on memory performance in the Results subsection “Behavioural Results”. Specifically, associative *d’* for word-face pairs (*M* = 2.16, *SD* = .68) was significantly higher than for word-place pairs (*M* = 1.85, *SD* = .74; *t*(99) = 5.37, *p* < 10^-7^). We note here that this behavioural difference did not have parallels in the neural data. In particular, mean reinstatement strength did not differ between face and place trials in ANG (place: M = 73.3%; face: M = 71.3%, *t*(99) = 1.69, *p* = .094), and was actually greater on place trials in VTC (place: M = 71.5%; face: M = 65.1%; *t*(99) = 5.25, *p* < 10^-7^). We now report on the effect of image category on reinstatement strength in ANG and VTC in the main manuscript (Results subsection “Memory Behaviour Scales with Trial-wise Category-level Reinstatement”).

b) Prior to each of 10 iterations of encoding classifier training, the data were subsampled to ensure an equal number of face and place study trials following exclusion of trials with artefacts (see Materials and methods subsection – “Multivoxel Pattern Classification”). This ensured that the classifier was unbiased with respect to image category, and therefore all classification outcomes cannot be confounded with image category.

c) We note that stimulus category was included as a regressor in all linear and logistic mixed effects models, which is now explicitly noted in the Results subsection “Memory Behaviour Scales with Trial-wise Category-level Reinstatement “ and made more evident by the listing of all predictors in mixed effects models in Supplementary file 1C. Importantly, by including category as a regressor, this further ensures that our key findings are not confounded with category, and instead generalize across category.

d) While the main text of the manuscript focuses on main effects, as these are central to our specific aims, we point readers to Supplementary file 1E-F within Results reporting (subsection “Memory Behaviour Scales with Trial-wise Category-level Reinstatement”) to obtain further details on the moderating effect of stimulus category on the key findings.

11) Please clarify how the fMRI measures were summarized for the individual difference analyses. Were they simply averaged across all trials, AH trials, or some other mixture? Also, why were age interactions excluded from the model? While the models suggest that the effects are independent of age, they do not imply that they are age invariant (Rugg, 2015). I appreciate that there might not be enough variability in this sample (which is only older adults) but demonstrating age invariance here seems important given the conclusions that are drawn.

We thank the reviewers for the opportunity to clarify these points. To address the first point, fMRI measures for the individual difference analyses reflect the strength of each measure during associative hit trials. In particular, category reinstatement strength in VTC and ANG reflects mean classifier evidence (i.e., mean logits) across all associative hit trials, event-level reinstatement strength in VTC and ANG reflects mean ERS on associative hit trials, and hippocampal activity reflects the difference between mean activity during associative hit trials and mean activity during correct rejections. We revised the manuscript in multiple locations (Results subsection “Effects of Age on Hippocampal and Cortical Indices of Pattern Completion”; Materials and methods subsections “Multivoxel Pattern Classification” and “Statistical Analysis”) to enhance clarity regarding these measures.

To address the second point, we did not initially include interactions with age in these models because we were interested in the relationship between the neural variables and memory performance, independent of age. As advised by the reviewers, we conducted analyses examining the interaction between each of our neural variables and age. These models did not demonstrate significant interactions with age when associative d’ was the dependent variable (hippocampal activity: *β* = -0.14, *p* = .133; category reinstatement strength in VTC: *β* = 0.09, *p* = .332; or ANG: *β* = -0.01, *p* = .951), nor when exemplar-specific recall was the dependent variable (hippocampal activity: *β* = -0.15, *p* = .088; category reinstatement strength in VTC: *β* = 0.003, *p* = .977 or ANG: *β* = 0.07, *p* = .565), suggesting the present results are both independent of age and age-invariant. However, as the reviewers suggest, we interpret these results with caution given that the present sample is selective to older adults. We now report these outcomes in the manuscript, including this important caveat (Results subsection “Neural Indices of Pattern Completion Explain Individual Differences in Episodic Memory”).

12) The degree of variability in any given sample/study will in part be driven by the recruitment procedure. It is our impression that this sample was very healthy and cognitively intact (see Table 1) – and there was no relation between level of education and memory performance! This is at odds with findings in many published longitudinal studies. Comments on the sample in relation to the topic of heterogeneity would be informative.

It is true that the present sample was healthy and cognitively intact. This was by design, as we sought to understand neural factors driving individual differences in memory performance within the normal range (i.e., excluding both subjective and mild cognitive impairment). Accordingly, inclusion criteria included CDR of 0 and performance within the normal range on a neuropsychological test battery, which necessarily restricted heterogeneity in memory performance. We recruited widely in the bay area through ads in newspapers, newsletters, and online (e.g., Facebook), with the following inclusion criteria: native English speaking, right-handed, and MRI compatible. We note that this convenience sample, together with the inclusion criteria and self-selection bias (characteristics of individuals who are able and motivated to come in for three study visits including two MRI scans and a lumbar puncture), mean that our sample is not representative of the wider population with respect to education, SES, and race/ethnicity. However, we note that despite these criteria, we still observe considerable heterogeneity in memory performance within this group, enabling us to address questions regarding individual differences in memory performance within the ‘cognitively normal’ range. The absence of a relationship between education and memory performance in our sample could be due to high educational attainment of participants on average, or the cross-sectional nature of the study design. We now comment on this in the revised manuscript (Discussion, last paragraph).

13) Effects now attributed to influences of ageing in this cross-sectional study could be driven by other factors, and strong conclusions on the effects go ageing will have to await longitudinal follow-up studies. Please address this limitation in the paper.

We appreciate the limitations of cross-sectional study designs in attributing observed age effects to chronological age per se. We have revised the manuscript to include consideration of this point (Discussion, last paragraph), including the necessity for longitudinal data in order to make strong claims regarding the effects of chronological age on the functional and behavioural measures reported here.

[Editors' note: further revisions were suggested prior to acceptance, as described below.]

The manuscript has been improved but there are some remaining issues that need to be addressed before acceptance, as outlined below:1) Please clarify throughout the paper, figures (i.e., axis labels), and legends when the term "reinstatement strength" refers to category-level versus event-level reinstatement.

We appreciate the importance of distinguishing between which measure of cortical reinstatement we are referring to throughout the manuscript, figures, and tables. We have revised the manuscript text throughout the Results, Discussion and Materials and methods, as well as the figures axes and figure legends (Figure 4 and Figure 4—figure supplements 1-6, Figure 5 and Figure 5—figure supplements 1-5), and tables and table captions (Table 2; Supplementary file 1C-H) to specify ‘event-level’ or ‘category-level’ reinstatement.

2) Please clarify what is meant by "univariate activity" in Supplementary file 1C. Are all univariate measures derived from hippocampus, or was activity from other ROIs (VTC, ANG) also included in some models as a control? Furthermore, what is the reason for sometimes using category-selective voxels and other times using whole-ROI activity? These details could be noted in the paper or table caption.

We thank you for pointing out the need for enhanced clarity on this point. In Supplementary file 1C, ‘Univariate Activity’ refers to activity in the target ROIs (VTC, ANG), whereas only ‘Hippocampal Activity’ refers to univariate activity in the hippocampus. The rationale for using different univariate activity control variables for the models containing classification metrics as compared to those containing pattern similarity metrics stems from differences in the analytic approach. In particular, classification analyses were conducted over category-selective voxels identified in a feature selection step, as is conventional for MVPA. In contrast, pattern similarity analyses were conducted over all voxels in the ROI. Thus, we control for univariate activity in the voxels that were submitted to the analysis in each case.

We added further description of these control variables and the corresponding rationale in the main manuscript (Materials and methods subsection “Statistical Analysis”). We additionally revised the terminology in Supplementary file 1C to ‘ROI Univariate Activity’, and have added further description of this term and the rationale for different control variables in the table caption as follows:

“^a^ = Univariate activity in top 500 category-selective voxels in VTC or ANG over which classification analyses were conducted; ^b^ = Univariate activity in whole ROI (VTC or ANG) over which pattern similarity analyses were conducted.”